# LEARNING TO CROSS EXCHANGE TO SOLVE MIN-MAX VEHICLE ROUTING PROBLEMS

**Minjun Kim**[*]**, Junyoung Park**[*]**, & Jinkyoo Park**
KAIST, Daejeon, South Korea
{minjun1212,junyoungpark,jinkyoo.park}@kaist.ac.kr

## ABSTRACT

CROSS exchange (CE), a meta-heuristic that solves various vehicle routing problems (VRPs), improves the solutions of VRPs by swapping the sub-tours of the vehicles. Inspired by CE, we propose Neuro CE (NCE), a fundamental operator of *learned* meta-heuristic, to solve various min-max VRPs while overcoming the limitations of CE, i.e., the expensive $\mathcal{O}(n^4)$ search cost. NCE employs graph neural network to predict the cost-decrements (i.e., results of CE searches) and utilizes the predicted cost-decrements to guide the selection of sub-tours for swapping, while reducing the search cost to $\mathcal{O}(n^2)$. As the learning objective of NCE is to predict the cost-decrement, the training can be simply done in a supervised fashion, whose training samples can be easily collected. Despite the simplicity of NCE, numerical results show that the NCE trained with min-max flexible multi-depot VRP (min-max FMDVRP) outperforms the meta-heuristic baselines. More importantly, it significantly outperforms the neural baselines when solving distinctive special cases of min-max FMDVRP (e.g., min-max MDVRP, min-max mTSP, min-max CVRP) without additional training.

## 1 INTRODUCTION

The field of neural combinatorial optimization (NCO), an emerging research area intersecting operation research and artificial intelligence, aims to train an effective solver for various combinatorial optimization (CO) problems, such as the traveling salesman problem (TSP) (Bello et al., 2016; Khalil et al., 2017; Nazari et al., 2018; Kool et al., 2018; Kwon et al., 2020), vehicle routing problems (VRPs) (Bello et al., 2016; Khalil et al., 2017; Nazari et al., 2018; Kool et al., 2018; Kwon et al., 2020; Hottung & Tierney, 2019; Lu et al., 2019; da Costa et al., 2021), and vertex covering problems (Khalil et al., 2017; Li et al., 2018; Guo et al., 2019). As NCO tackles NP-hard problems using various state-of-the-art (SOTA) deep learning techniques, it is considered an important research area in artificial intelligence. At the same time, NCO is an important field from a practical point of view because it can solve complex real-world problems. The current study mainly focuses on VRPs, a type of CO problems.

Majority of learning-based VRP solvers learns to improve the current solution to obtain a better solution (i.e., improvement heuristics) (Hottung & Tierney, 2019; Lu et al., 2019; da Costa et al., 2021) or construct a solution sequentially (i.e., construction heuristics) (Bello et al., 2016; Khalil et al., 2017; Nazari et al., 2018; Kool et al., 2018; Kwon et al., 2020; Park et al., 2021; Cao et al., 2021). To learn such solvers, learning-based methods either employ supervised learning (SL), which imitates the solutions of the verified solvers, or reinforcement learning (RL), which learn a solver from the generated routes. Most NCO studies focus on the well-established "min-sum VRP" that aims to minimize the total traveling distance of vehicles, possibly because the benchmark problems and baseline algorithms are set up for the "min-sum VRP." On the other hand, VRP with different objectives have not received much attention from the NCO community, even though they can model various practical scenarios. For example, "min-max VRP" that aims to minimize the total completion time (i.e., makespan) of various time-critical distributed tasks (e.g., vaccine delivery, grocery delivery) has not been widely considered.

---

[*]Equal contribution

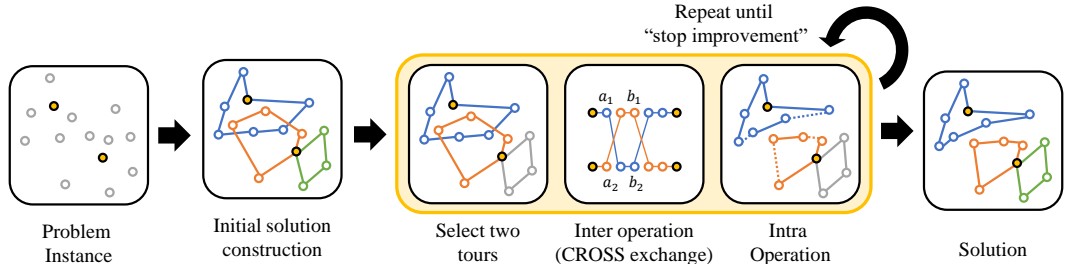

Figure 1: The overall procedure of improvement heuristic that uses CE as the inter-operation.

This study aims to learn a fundamental and universal operator that can effectively solve various practical "min-max VRP" that have flexible depot constraints. To design an universal and simple, yet powerful operator, we utilize CROSS-exchange (CE) (Taillard et al., 1997), a local search designed to conduct the inter-operation of two routes (i.e., swapping the sub-tours of two selected routes) to reduce the traveling cost. We noticed that the inter-operation of CE is especially effective in improving the quality of "min-max VRP" because it can consider the interaction among multiple vehicles, and effectively reduce the differences between the traveling distances of all vehicles. However, the search cost for selecting the sub-tours from two trajectory is $\mathcal{O}(n^4)$ where $n$ is the tour length. This make CE unapplicable to large scale VRPs.

In this paper, we propose Neuro CE (NCE) that effectively conducts the CE operation with significantly less computational complexity. NCE amortizes the search for ending nodes of the sub-tours by employing a graph neural network (GNN), which predicts the best cost decrement, given two starting nodes from two given trajectories. NCE searches over only promising starting nodes using the prediction. Hence, the proposed NCE has $\mathcal{O}(n^2)$ search complexity. Furthermore, unlike other SL or RL approaches, the prediction target labels of NCE is not the entire solution of the VRPs, but the cost decrements of the CE operations (true target operator), which makes the collection of training data simple and easy.

The contributions of this study are summarized as follows:

- **Generalizability/Transferability:** As NCE learns a fundamental and universal operator to solve various complex min-max VRPs without retraining for each type of VRPs.
- **Trainability:** The NCE operator is trained in a supervised manner with the dataset comprised of the tour pairs and cost decrements, which make the collection of training data easy and simple.
- **Practicality/Performance:** We evaluate NCE with various types of min-max VRPs, including flexible multi-depot VRP (min-max FMDVRP), multi-depot VRP (min-max MDVRP), multiple TSP (min-max mTSP), and capacitated VRP (min-max CVRP). Extensive numerical experiments validate that NCE outperforms SOTA meta-heuristics and NCO baselines in solving various min-max VRPs, even though NCE is only trained with the data from min-max FMDVRP.

## 2 PRELIMINARIES

This section introduces the target problem, min-max flexible multi-depot VRP (min-max FMD-VRP), and the CE operator, a powerful local-search heuristics that solves min-max FMDVRP.

### 2.1 MIN-MAX FLEXIBLE MULTI-DEPOT VRP

Min-max FMDVRP is a generalization of VRP that aims to find the coordinated routes of multiple vehicles with multiple depots. The flexibility allows vehicles to go back to any depots regardless of their starting depots. The min-max FMDVRP is formulated as follows:

$$\min_{\pi \in \mathbb{S}(P)} \max_{i \in \mathbb{V}} C(\tau_i) \tag{1}$$

where $P$ is the description of the min-max FMDVRP instance that is composed of a set of vehicles $\mathbb{V}$, $\mathbb{S}(P)$ is the set of solutions that satisfy the constraints of the min-max FMDVRP (i.e., feasible solutions), and $\pi = \{\tau_i\}_{i \in \mathbb{V}}$ is a solution of the min-max FMDVRP. The tour $\tau_i = [N_1, N_2, ..., N_{l(i)}]$

---

**Algorithm 1:** Neuro CROSS exchange (NCE) for solving VRP family

---

**Input:** VRP instance $P$, cost-decrement prediction model $f_\theta$, Perturbation parameter $p$
**Output:** Optimized tours $\{\tau_i^*\}_{i\in|\mathbb{V}|}$

1   $\{\tau_i\}_{i\in|\mathbb{V}|} \leftarrow \text{GetInitialSolution}(P)$
2   $C_{\text{per}} \leftarrow 0$
3   **while** *True* **do**
4      **while** *improvement* **do**
5          $(\tau_1, \tau_2) \leftarrow \text{SelectTours}(\{\tau_i\}_{i\in|\mathbb{V}|})$
6          $(\tau_1', \tau_2') \leftarrow \text{NeuroCROSS}(\tau_1, \tau_2, f_\theta)$                   `// Inter operation`
7          $\tau_i' \leftarrow \text{IntraOperation}(\tau_i), i = 1, 2$
8          $\tau_1 \leftarrow \tau_1', \tau_2 \leftarrow \tau_2'$
9      **if** *update* **then**
10          $\{\tau_i^*\}_{i\in|\mathbb{V}|} \leftarrow \{\tau_i\}_{i\in|\mathbb{V}|}$
11      **if** $C_{per} = p$ **then**
12          **break**
13      $C_{\text{per}} \leftarrow C_{\text{per}} + 1$
14      $(\tau_1, \tau_2) \leftarrow \text{ChooseRandomTours}$
15      $(\tau_1, \tau_2) \leftarrow \text{RandomExchange}(\tau_1, \tau_2)$             `// escape from local minima`

---

of vehicle $i$ is the ordered collection of the visited tasks by the vehicle $v_i$, and $C(\tau_i)$ is the cost of $\tau_i$. Min-max FMDVRP can be used to formulate the operation of shared vehicles that can be picked up from or delivered to any depots. The mixed integer linear programming (MILP) formulation of min-max FMDVRP is provided in Appendix A.3.

Classical VRPs are special cases of FMDVRP: *TSP* is a VRP with a single vehicle and depot, *mTSP* is a VRP with multiple vehicles and a single depot, and *MDVRP* is a VRP with multiple vehicles and depots. Since FMVDRP is a general problem class, we learn a solver for FMVDRP and employ it to solve other specific problems (i.e., min-max MDVRP, min-max mTSP, and min-max CVRP), without retraining or fine-tuning. We demonstrate that the proposed method can solve these special cases almost optimally without retraining in Section 4.

## 2.2 CROSS EXCHANGE

CE is a solution updating operator that iteratively improves the solution until it reaches a satisfactory result (Taillard et al., 1997). CE reduces the overall cost by *exchanging* the sub-tours in two tours. The CE operator is defined as:

$$\tau_1', \tau_2' = \text{CROSS}(a_1, b_1, a_2, b_2; \tau_1, \tau_2) \tag{2}$$

$$\tau_1' \triangleq \tau_1[: a_1] \oplus \tau_2[a_2 : b_2] \oplus \tau_1[b_1 :] \tag{3}$$

$$\tau_2' \triangleq \tau_2[: a_2] \oplus \tau_1[a_1 : b_1] \oplus \tau_2[b_2 :] \tag{4}$$

where $\tau_i$ and $\tau_i'$ are the input and updated tours of the vehicle $i$, respectively. $\tau_i[a : b]$ represents the sub-tour of $\tau_i$, ranging from node $a$ to $b$. $\tau \oplus \tau'$ represents the concatenation of tours $\tau$ and $\tau'$. For brevity, we assume that node $a_1, a_2$ comes early than node $b_1, b_2$ in $\tau_1, \tau_2$, respectively.

CE selects the sub-tours (i.e., $\tau_1[a_1 : b_1], \tau_2[a_2 : b_2]$) from $\tau_1, \tau_2$ and swaps the sub-tours to generate new tours $\tau_1', \tau_2'$. CE seeks to find the four points $(a_1, b_1, a_2, b_2)$ to reduce the cost of the tours. For min-max VRPs, we define the cost of the two selected tours as $C(\tau_1, \tau_2) = \max(l(\tau_1), l(\tau_2))$, where $l(\tau_i)$ is the traveling distance of tour $\tau_i$, and apply the CE operator to reduce this cost, i.e., $C(\tau_1', \tau_2') \leq C(\tau_1, \tau_2)$ in an attempt to minimize the traveling distance of the longest route. When the full search method is naively employed, the search cost is $\mathcal{O}(n^4)$, where $n$ is the number of nodes in a tour.

Fig. 1 illustrates how the improvement heuristics utilize CE to solve min-max FMDVRP. The improvement heuristics start by generating the initial feasible tours using simple heuristics. Then, they repeatedly (1) select two tours, (2) apply inter-operation to generate improved tours by CE, and (3) apply intra-operation to improve the tours independently. The improvement heuristics terminate when no more (local) improvement is possible.

---

**Algorithm 2:** `NeuroCROSS` operation

---

**Input:** tours $\tau_1, \tau_2$, cost-decrement prediction model $f_\theta(\cdot)$
**Output:** updated tours $\tau_1', \tau_2'$
    /* Predict cost decrement                                                     */
16   $\mathbb{S} \leftarrow \{\emptyset\}$
17 **for** $(a_1, a_2) \in \tau_1 \times \tau_2$ **do**
18     $\hat{y}^*(a_1, a_2; \tau_1, \tau_2) \leftarrow f_\theta(a_1, a_2; \tau_1, \tau_2)$             // Cost-decrement prediction
19     $\mathbb{S} \leftarrow \mathbb{S} \cup \{((a_1, a_2), \hat{y}^*(a_1, a_2; \tau_1, \tau_2))\}$
    /* Candidate set construction                                           */
20 Sort $\mathbb{S}$ by $y^*(a_1, a_2; \tau_1, \tau_2)$ in the descending order
21 $\mathbb{S}_K \leftarrow$ Take first $K$ elements of $\mathbb{S}$
    /* Perform search                                                          */
22 $a_1^* \leftarrow \emptyset, a_2^* \leftarrow \emptyset, b_1^* \leftarrow \emptyset, b_2^* \leftarrow \emptyset, y^* \leftarrow 0$
23 **for** $((a_1, a_2), \hat{y}^*(a_1, a_2; \tau_1, \tau_2)) \in \mathbb{S}_K$ **do**
24     $(\bar{b}_1, \bar{b}_2) \leftarrow \arg\max_{b_1, b_2} (C(\text{CROSS}((a_1, b_1, a_2, b_2; \tau_1, \tau_2))) - C(\tau_1, \tau_2))$
25     $y^*(a_1, a_2; \tau_1, \tau_2) \leftarrow C(\text{CROSS}((a_1, \bar{b}_1, a_2, \bar{b}_2; \tau_1, \tau_2)) - C(\tau_1, \tau_2)$
26     **if** $y^*(a_1, a_2; \tau_1, \tau_2) \geq y^*$ **then**
27        $a_1^* \leftarrow a_1, a_2^* \leftarrow a_2, b_1^* \leftarrow \bar{b}_1, b_2^* \leftarrow \bar{b}_2$
28        $y^* \leftarrow y^*(a_1, a_2; \tau_1, \tau_2)$
29 $(\tau_1', \tau_2') \leftarrow \text{CROSS}(a_1^*, b_1^*, a_2^*, b_2^*; \tau_1, \tau_2)$

---

## 3   NEURO CROSS EXCHANGE

In this section, we introduce Neuro CROSS exchange (NCE) to solve min-max FMDVRP and its special cases. The overall procedure of NCE is summarized in Algorithm 1. We briefly explain `GetInitialSolution`, `SelectTours`, `NeuroCROSS`, and `IntraOperation`, and then provide the details of the proposed `NeuroCROSS` operation in the following subsections. NCE is particularly designed to improve the solution quality and problem solving speed of CE when solving min-max VRPs. Each component of NCE is as follows:

- **`GetInitialSolution`.** We use a multi-agent extended version of the greedy assignment heuristic (Dell'Amico et al., 1993) to obtain the initial feasible solutions. The heuristic first clusters the cities into $|\mathbb{V}|$ clusters and then applies the greedy assignment to each cluster to get the initial solution.

- **`SelectTours`.** Following the common practice, we select $\tau_1, \tau_2$ as the tours with the largest and smallest traveling distance (i.e., $\tau_1 = \arg\max_\tau(l(\tau_i)_{i \in \mathbb{V}}), \tau_2 = \arg\min_\tau(l(\tau_i)_{i \in \mathbb{V}})$).

- **`NeruoCROSS`.** We utilize the cost-decrement prediction model $f_\theta(\cdot)$ and two-stage search method to find the cost-improving tour pair $(\tau_1', \tau_2')$ with $\mathcal{O}(n^2)$ budget. The details of the NCE operation will be given in Sections 3.1 and 3.2.

- **`IntraOperation`.** For our target VRPs, the intra-operation is equivalent to solving TSP. We utilize `elkai` (Dimitrovski) to solve TSP.

### 3.1   NEURO CROSS EXCHANGE OPERATION

The CE operation can be shown as selecting two pairs of nodes (i.e., the pairs of $a_1/b_1$ and $a_2/b_2$) from the selected tours (i.e., $\tau_1, \tau_2$). This typically involves $\mathcal{O}(n^4)$ searches. To reduce the high search complexity, NCE utilizes the cost-decrement model $f_\theta(a_1, a_2; \tau_1, \tau_2)$ that predicts the maximum cost decrements from the given $\tau_1$ and $\tau_2$, and the starting nodes $a_1$ and $a_2$ of their sub-tours. That is, $f_\theta(a_1, a_2; \tau_1, \tau_2)$ amortizes the search for the ending nodes $b_1, b_2$ given $(\tau_1, \tau_2, a_1, a_2)$, and helps to identify the promising $(a_1, a_2)$ pairs that are likely to improve the tours. After selecting the top $K$ promising pairs of $(a_1, a_2)$ using $f_\theta(a_1, a_2; \tau_1, \tau_2)$, whose search cost is $\mathcal{O}(n^2)$, NCE then finds $(b_1, b_2)$ to identify the promising $(a_1, a_2)$ pairs. Overall, the entire search can be done in $\mathcal{O}(n^2)$. The following paragraphs detail the procedures of NCE.

**Predicting cost decrement.** We employ $f_\theta(a_1, a_2; \tau_1, \tau_2)$ (which will be explained in Section 3.2) to predict the optimal cost decrement $y^*$ defined as:

$$y^*(a_1, a_2; \tau_1, \tau_2) = \max_{b_1, b_2} \left( C(\text{CROSS}((a_1, b_1, a_2, b_2; \tau_1, \tau_2))) - C(\tau_1, \tau_2) \right) \quad (5)$$

$$\approx f_\theta(a_1, a_2; \tau_1, \tau_2) \quad (6)$$

where $C(\tau_1, \tau_2) = \max(l(\tau_1), l(\tau_2))$. In other words, $f_\theta(\cdot)$ predicts the best cost decrement of $\tau_1$ and $\tau_2$, given $a_1$ and $a_2$. The real cost decrement labels are obtained from the real search operation.

**Constructing search candidate set.** By training $f_\theta(\cdot)$, we can amortize the search for $b_1$ and $b_2$. However, this amortization bears the prediction errors, which can misguide entire improvement process. To alleviate this problem, we selects the top $K$ pairs of $(a_1, a_2)$ that have the top $K$-largest $y^*$ out of all $(a_1, a_2)$ choices. Intuitively speaking, NCE exclude the less promising $(a_1, a_2)$ pairs while utilizing the prediction model $f_\theta(\cdot)$ which can have some errors.

**Performing reduced search.** NCE finds the best $(b_1, b_2)$ for each $(a_1, a_2)$ in the search candidate sets and selects the best sub-tours $(a_1, a_2, b_1, b_2)$ that maximizes the actual cost decrement (not prediction). Unlike the full search of CE, the proposed NCE only performs the search for $(b_1, b_2)$, which reduces the search cost from $\mathcal{O}(n^4)$ to $\mathcal{O}(n^2)$. The detailed procedures of NCE are summarized in Algorithm 2.

## 3.2 Cost-decrement prediction model

NCE saves computations by employing $f_\theta(a_1, a_2; \tau_1, \tau_2)$ to predict $y^*(\cdot)$ from $a_1, a_2, \tau_1$ and $\tau_2$. The overall procedure is illustrated in Fig. 2.

**Graph representation of** $(\tau_1, \tau_2)$. We represent a pair of tours $(\tau_1, \tau_2)$ as a directed complete graph $\mathcal{G} = (\mathbb{N}, \mathbb{E})$, where $\mathbb{N} = \tau_1 \cup \tau_2$ (i.e., the $i^{\text{th}}$ node $n_i$ of $\mathcal{G}$ is either the city or depot of the tours, and $e_{ij}$ is the edge from $n_i$ to $n_j$). $\mathcal{G}$ has the following node and edge features:

- $x_i = [\textbf{coord}(n_i), \mathbb{1}_{\textbf{depot}}(n_i)]$, where $\textbf{coord}(n_i)$ is the 2D Euclidean coordinate of $v_i$, and $\mathbb{1}_{\textbf{depot}}(n_i)$ is the indicator of whether $n_i$ is a depot.
- $x_{ij} = [\textbf{dist}(n_i, n_j)]$, where $\textbf{dist}(n_i, n_j)$ is the 2D Euclidean distance between $n_i$ and $n_j$.

**Graph embedding with attentive graph neural network.** We employ an attentive variant of graph-network (GN) block (Battaglia et al., 2018) to embed $\mathcal{G}$. The attentive embedding layer is defined as follows:

$$h'_{ij} = \phi_e(h_i, h_j, h_{ij}, x_{ij}) \quad (7)$$

$$z_{ij} = \phi_w(h_i, h_j, h_{ij}, x_{ij}) \quad (8)$$

$$w_{ij} = \text{softmax}(\{z_{ij}\}_{j \in \mathcal{N}(i)}) \quad (9)$$

$$h'_i = \phi_n(h_i, \sum_{j \in \mathcal{N}(i)} w_{ij} h'_{ij}) \quad (10)$$

where $h_i$ and $h_{ij}$ are node and edge embeddings respectively, $\phi_e$, $\phi_w$, and $\phi_n$ are the Multilayer Perceptron (MLP)-parameterized edge, attention and node operators respectively, and $\mathcal{N}(i)$ is the neighbor set of $n_i$. We utilize $H$ embedding layers to compute the final node $\{h_i^{(H)} | n_i \in \mathbb{V}\}$ and edge $\{h_{ij}^{(H)} | e_{ij} \in \mathbb{E}\}$ embeddings.

**Cost-decrement prediction.** Based on the computed embedding, the cost prediction module $\phi_c$ predicts $y^*(a_1, a_2; \tau_1, \tau_2)$. The selection of the two starting nodes in $\tau_1$ and $\tau_2$ indicates (1) the addition of the two edges, $(a_1, a_2 + 1)$ and $(a_2, a_1 + 1)$, and (2) the removal of the original two edges, $(a_1, a_1 + 1)$ and $(a_2, a_2 + 1)$, as shown in the third block in Fig. 2 (we overload the notations $a_1 + 1, a_2 + 1$ so that they denote the next nodes of $a_1, a_2$ in $\tau_1, \tau_2$, respectively). To consider such edge addition and removal procedure in cost prediction, we design $\phi_c$ as follows:

$$\hat{y}^*(a_1, a_2; \tau_1, \tau_2) = \phi_c(\underbrace{h_{a_1}^{(H)}, h_{a_1+1}^{(H)}, h_{a_2}^{(H)}, h_{a_2+1}^{(H)}}, \underbrace{h_{a_1,a_2+1}^{(H)}, h_{a_2,a_1+1}^{(H)}}, \underbrace{h_{a_1,a_1+1}^{(H)}, h_{a_2,a_2+1}^{(H)}}) \quad (11)$$

● / ● : node embedding  —— : link addition  ····· : link removal

Figure 2: Cost-decrement prediction procedure

where $h_i^{(H)}$ and $h_{i,j}^{(H)}$ denotes the embedding of $n_i$ and $e_{ij}$, respectively.

The quality of the NCE operator highly depends on the accuracy of $f_\theta$. When $K \geq 10$, we experimentally confirmed that the NCE operator finds the argmax $(a_1, a_2, b_1, b_2)$ pair with high probability. We provide the experimental details and results about the predictions of $f_\theta$ in Appendix H.

## 4 EXPERIMENTS

This section provides the experiment results that validate the effectiveness of the proposed NCE in solving min-max FMDVRP and various min-max VRPs. To train $f_\theta(\cdot)$, we use the input $(\tau_1, \tau_2, a_1, a_2)$ and output $y^*$ pairs obtained from 50,000 random min-max FMDVRP instances. The details of the train data generation are described in Appendix G. The cost decrement model $f_\theta(\cdot)$ is parametrized by the GNN that contains the five attentive embedding layers. The details of the $f_\theta(\cdot)$ architecture and the computing infrastructure used to train $f_\theta(\cdot)$ are discussed in Appendix G.

We emphasize that we use a single $f_\theta(\cdot)$ that is trained using data obtained from random min-max FMDVRP instances for all experiments. We found that $f_\theta(\cdot)$ effectively solves the three special cases (i.e., min-max MDVRP, min-max mTSP, and min-max CVRP) without retraining or fine-tuning, proving the effectiveness of NCE as a universal operator for VRPs.

### 4.1 MIN-MAX FMDVRP EXPERIMENTS

We evaluate the performance of NCE in solving various sizes of min-max FMDVRP. We consider 100 random min-max FMDVRP instances for each problem size $(N_c, N_d, N_v)$, where $N_c, N_d, N_v$ are the number of cities, depots, and vehicles, respectively. We provide the average makespan and computation time for the 100 instances. For small-sized problems ($N_c \leq 10$), we employ CPLEX (Cplex, 2009) (an exact method), OR-tools (Perron & Furnon), CE (full search), ScheduleNet (Park et al., 2021), greedy heuristic (Dell'Amico et al., 1993), and greedy + TSP heuristic as baselines. For the larger-sized problems, we exclude CPLEX from the baselines due to its limited scalability. To our knowledge, our method is the first neural approach to solve min-max FMDVRP. Please note that we extend the ScheduleNet algorithm (Park et al., 2021), the most performing neural baseline for mTSP, and utilize it as a neural baseline for the min-max FMVDRP experiments.

Table 1: **FMDVRP results** (small-sized instances)

| $N_c, N_d$ ($\downarrow$) | $N_v (\rightarrow)$ Method | 2 Cost | Gap(%) | Time(sec.) | 3 Cost | Gap(%) | Time(sec.) |
|---|---|---|---|---|---|---|---|
| | CPLEX | **1.543** | 0.00 | 0.31 | **1.363** | 0.00 | 0.83 |
| (7,2) | OR-tools | 1.596 | 3.43 | 0.01 | 1.380 | 1.25 | 0.01 |
| | CE | 1.546 | 0.02 | 0.04 | 1.364 | 0.01 | 0.03 |
| | NCE | 1.546 | 0.02 | 0.10 | 1.365 | 0.01 | 0.12 |
| | $N_v (\rightarrow)$ Method | 2 Cost | Gap(%) | Time(sec.) | 3 Cost | Gap(%) | Time(sec.) |
| | CPLEX | **1.745** | 0.00 | 9.29 | **1.488** | 0.00 | 63.00 |
| (10,2) | OR-tools | 1.820 | 4.30 | 0.02 | 1.521 | 2.22 | 0.02 |
| | CE | 1.749 | 0.02 | 0.07 | 1.493 | 0.03 | 0.06 |
| | NCE | 1.749 | 0.02 | 0.13 | 1.493 | 0.03 | 0.16 |

Table 2: **FMDVRP results** (medium-sized instances)

| $N_c, N_d$ ($\downarrow$) | $N_v(\rightarrow)$ Method | 3 Cost | Gap(%) | Time(sec.) | 5 Cost | Gap(%) | Time(sec.) | 7 Cost | Gap(%) | Time(sec.) |
|---|---|---|---|---|---|---|---|---|---|---|
| | OR-tools | 2.39 | 15.46 | 2.20 | 1.56 | 10.64 | 2.44 | 1.27 | 6.72 | 2.58 |
| (50,6) | ScheduleNet | 2.61 | 26.09 | 5.21 | 1.86 | 31.91 | 5.44 | 1.57 | 31.93 | 6.14 |
| | Greedy | 3.01 | 46.38 | 0.01 | 2.24 | 58.87 | 0.01 | 1.99 | 67.23 | 0.01 |
| | Greedy + TSP | 2.75 | 32.85 | 0.03 | 2.12 | 50.35 | 0.02 | 1.91 | 60.50 | 0.02 |
| | CE | **2.07** | 0.00 | 21.06 | 1.41 | 0.00 | 9.09 | 1.19 | 0.00 | 5.37 |
| | NCE | 2.08 | 0.48 | 1.26 | **1.40** | -0.71 | 1.82 | **1.19** | 0.00 | 2.23 |
| | $N_v(\rightarrow)$ Method | 5 Cost | Gap(%) | Time(sec.) | 7 Cost | Gap(%) | Time(sec.) | 10 Cost | Gap(%) | Time(sec.) |
| | OR-tools | 2.00 | 14.94 | 30.46 | 1.51 | 12.69 | 32.25 | 1.20 | 10.09 | 34.38 |
| (100,8) | ScheduleNet | 2.32 | 33.33 | 35.47 | 1.86 | 38.81 | 36.08 | 1.54 | 41.28 | 41.30 |
| | Greedy | 2.82 | 62.07 | 0.04 | 2.37 | 76.87 | 0.05 | 2.00 | 83.49 | 0.05 |
| | Greedy + TSP | 2.60 | 49.43 | 0.07 | 2.25 | 67.16 | 0.07 | 1.92 | 76.16 | 0.07 |
| | CE | **1.74** | 0.00 | 218.46 | 1.34 | 0.00 | 128.40 | 1.09 | 0.00 | 78.56 |
| | NCE | 1.75 | 0.57 | 6.41 | **1.34** | 0.00 | 9.54 | **1.09** | 0.00 | 13.34 |

Table 3: **FMDVRP results** (large-sized instances)

| $N_c, N_d, N_v(\rightarrow)$ Method | 200,10,10 Cost | Gap(%) | Time(sec.) | 400,20,20 Cost | Gap(%) | Time(sec.) | 600,30,30 Cost | Gap(%) | Time(sec.) | 800,40,40 Cost | Gap(%) | Time(sec.) |
|---|---|---|---|---|---|---|---|---|---|---|---|---|
| OR-tools | 1.47 | 12.21 | 418.8 | 1.08 | 12.50 | 9484 | 0.88 | 17.33 | 25967 | 1.09 | 67.69 | 72166 |
| ScheduleNet | 1.87 | 42.75 | 244.8 | 1.43 | 48.97 | 340 | 1.38 | 84.00 | 970 | 1.18 | 81.54 | 2383 |
| Greedy | 2.44 | 86.26 | 0.22 | 2.14 | 122.92 | 1.46 | 2.32 | 209.33 | 2.71 | 1.85 | 184.62 | 5.78 |
| Greedy + TSP | 2.28 | 74.05 | 0.32 | 2.06 | 114.58 | 1.54 | 2.20 | 193.33 | 2.90 | 1.75 | 169.23 | 6.05 |
| NCE | **1.31** | 0.00 | 70.96 | **0.96** | 0.00 | 520.5 | **0.75** | 0.00 | 1940 | **0.65** | 0.00 | 3802 |

Table 1 shows the performances of NCE on the small-sized problems. NCE achieve similar makespans with CPLEX (optimal solution) within significantly lower computation times. NCE outperforms OR-tools in terms of makespan but has longer computation time; however, the computation time for NCE will be much lower than that of OR-tools when the problem size becomes bigger. It is noteworthy that NCE exhibits larger computation time than CE as the forward-propagation cost of GNN is larger than exhaustive search for small problems.

Table 2 and Table 3 shows the performances of NCE on the medium and large-sized problems, respectively. Applying CPLEX for large min-max FMDVRPs is infeasible, so we exclude it from the baselines. Instead, the CE serves as an oracle to compute the makespans. For all cases, NCE provides a solution with almost zero gap from CE , but is computationally much faster. This validates that NCE successfully amortizes the search operations of CE with significantly lower computation times. In addition, NCE consistently outperforms OR-tools for both the makespan and computational time. The performance gap between NCE and OR-tools becomes more significant as $N_c/N_v$ becomes large (i.e., each tour length becomes longer).

**min-max MDVRP results.** We also apply NCE with $f_\theta$ that is trained on FMDVRP to solve min-max MDVRP. As shown Tables A.1 to A.3 in Appendix B, NCE shows leading performance and is faster than the baselines, similar to the FMDVRP experiments.

## 4.2 MIN-MAX MTSP EXPERIMENTS

We evaluate the generalization capability of NCE for solving min-max mTSPs. We provide the average performance of 100 instances for each $(N_c, N_v)$ pair. For the baselines, we consider two meta-heuristics; LKH-3 (Helsgaun, 2017), which is known as the one of the best mTSP heuristics, and OR-tools, and two neural baselines; ScheduleNet (Park et al., 2021) and DAN (Cao et al., 2021).

As shown in Table 4, NCE achieves similar performance with LKH-3 with significantly shorter computational time. It is noteworthy that LKH-3 employs mTSP-specific heuristics on top of LKH heuristics, while NCE does not employ any mTSP-specific structures. To validate the effect of task-specific information on NCE, we train NCE with mTSP data (NCE-mTSP) and solve mTSP. The performances of NCE and NCE-mTSP are almost identical, which indicates that NCE is highly generalizable. In addition, NCE consistently outperforms the neural baselines. We further apply

Table 4: **Average makespans of the random mTSPs:** DAN and ScheduleNet results are taken from the original papers, † computational time of DAN is measured with the Nvidia RTX 3090.

| $N_c$ ($\downarrow$) | $N_v$ ($\rightarrow$) Method | 5 Cost | Gap(%) | Time(sec.) | 7 Cost | Gap(%) | Time(sec.) | 10 Cost | Gap(%) | Time(sec.) |
|---|---|---|---|---|---|---|---|---|---|---|
|  | LKH-3 | **2.00** | 0.00 | 187.46 | **1.95** | 0.00 | 249.31 | 1.91 | 0.00 | 170.20 |
|  | OR-tools | 2.04 | 2.00 | 3.24 | 1.96 | 0.51 | 3.75 | 1.91 | 0.00 | 3.67 |
| 50 | DAN | 2.29 | 14.50 | 0.25† | 2.11 | 8.21 | 0.26† | 2.03 | 6.28 | 0.30† |
|  | ScheuduleNet | 2.17 | 8.50 | 1.60 | 2.07 | 6.15 | 1.67 | 1.98 | 3.66 | 1.90 |
|  | NCE | 2.02 | 1.00 | 2.25 | 1.96 | 0.51 | 2.44 | **1.91** | 0.00 | 3.38 |
|  | NCE-mTSP | 2.02 | 1.00 | 2.48 | 1.96 | 0.51 | 2.50 | **1.91** | 0.00 | 3.44 |

| | $N_v$ ($\rightarrow$) Method | 5 Cost | Gap(%) | Time(sec.) | 10 Cost | Gap(%) | Time(sec.) | 15 Cost | Gap(%) | Time(sec.) |
|---|---|---|---|---|---|---|---|---|---|---|
|  | LKH-3 | **2.20** | 0.00 | 262.85 | 1.97 | 0.00 | 474.78 | 1.98 | 0.00 | 378.90 |
|  | OR-tools | 2.41 | 9.55 | 35.47 | 2.03 | 3.05 | 45.40 | 2.03 | 2.53 | 48.86 |
| 100 | DAN | 2.72 | 23.64 | 0.43† | 2.17 | 10.15 | 0.48† | 2.09 | 5.56 | 0.58† |
|  | ScheuduleNet | 2.59 | 17.73 | 14.84 | 2.13 | 8.12 | 16.22 | 2.07 | 4.55 | 20.02 |
|  | NCE | 2.25 | 2.27 | 16.01 | 1.98 | 0.51 | 12.22 | **1.98** | 0.00 | 24.08 |
|  | NCE-mTSP | 2.24 | 1.82 | 16.36 | **1.97** | 0.00 | 13.00 | **1.98** | 0.00 | 23.37 |

| | $N_v$ ($\rightarrow$) Method | 10 Cost | Gap(%) | Time(sec.) | 15 Cost | Gap(%) | Time(sec.) | 20 Cost | Gap(%) | Time(sec.) |
|---|---|---|---|---|---|---|---|---|---|---|
|  | LKH-3 | **2.04** | 0.00 | 1224.40 | 2,00 | 0.00 | 1147.13 | 1.97 | 0.00 | 908.14 |
|  | OR-tools | 2.33 | 14.22 | 675.79 | 2.33 | 16.50 | 604.31 | 2.37 | 20.30 | 649.17 |
| 200 | DAN | 2.40 | 17.65 | 0.93† | 2.20 | 10.00 | 0.98† | 2.15 | 9.14 | 1.07† |
|  | ScheuduleNet | 2.45 | 20.10 | 193.41 | 2.24 | 12.00 | 213.07 | 2.17 | 10.15 | 225.50 |
|  | NCE | 2.06 | 0.98 | 83.82 | **2.00** | 0.00 | 72.32 | **1.97** | 0.00 | 113.74 |
|  | NCE-mTSP | 2.06 | 0.98 | 84.96 | **2.00** | 0.00 | 84.28 | **1.97** | 0.00 | 118.55 |

NCE to solve mTSPLib mTSPLib, which comprises of mTSP instances from real cities, and large size min-max mTSPs. The experiment results are in Appendix C.

### 4.3 MIN-MAX CVRP EXPERIMENTS

Table 5: **min-max CVRP results:** $(s.n)$ indicates the best results of $n$ sampling. The baseline results are taken from (Bogyrbayeva et al., 2021).

| $N_c, N_v$ ($\rightarrow$) Method | 20,3 Cost | Gap(%) | Time(sec.) | 30,3 Cost | Gap(%) | Time(sec.) |
|---|---|---|---|---|---|---|
| OR-tools | 2.04 | 0.99 | 1.0 | 2.44 | 11.42 | 1.0 |
| AM (Bogyrbayeva et al., 2021) | 2.20 | 8.91 | 0.1 | 2.47 | 12.79 | 0.2 |
| AM (s.1200) (Bogyrbayeva et al., 2021) | 2.44 | 20.79 | 11.4 | 2.29 | 4.57 | 27.6 |
| HM (Bogyrbayeva et al., 2021) | 2.28 | 12.87 | 0.1 | 2.39 | 9.13 | 0.2 |
| HM (s.1200) (Bogyrbayeva et al., 2021) | 2.15 | 6.44 | 14.3 | 2.27 | 3.65 | 25.2 |
| NCE | 2.06 | 1.98 | 1.16 | 2.25 | 2.74 | 2.03 |
| NCE(s.10) | **2.02** | 0.00 | 2.31 | **2.19** | 0.00 | 5.78 |

We evaluate the generalization capability of NCE in solving min-max capacitated VRP. As $f_\theta(\cdot)$ is trained on min-max FMDVRPs, it does not consider the capacity constraints. However, we can easily enforce such constraints without retraining $f_\theta(\cdot)$, but by adjusting the searching range as follows:

$$(b_1, b_2) \leftarrow \underset{b_1, b_2 \in S_c}{\arg\max} \left( C(\text{CROSS}((a_1, b_1, a_2, b_2; \tau_1, \tau_2))) - C(\tau_1, \tau_2) \right), \qquad (12)$$

where the searching range $S_c$ is a set of nodes that satisfies the capacity constraints. As shown in Table 5, NCE outperforms other neural baselines (i.e., AM and HM) with shorter computation times, which again proves the effectiveness of NCE as a universal operator. Min-max CVRP with CVRPlib results are provided in Appendix D, Table A.6.

As most NCO studies use min-sum CVRP as the canonical benchmark tasks, we also employed NCE to solve min-sum CVRP problems. For min-sum CVRP, we trained the NCE cost decrement prediction model using different cost definition $C(\tau_1, \tau_2) = l(\tau_1) + l(\tau_2)$ in an attempt to minimize the

sum of the traveling distance. The min-sum CVRP benchmark results are provided in Appendix E, Table A.7.

### 4.4 ABLATION STUDIES

We evaluate the effects of the hyperparameters on NCE. The results are as follows:

- Appendix F.1: the performance of NCE converges when the number of candidate $K \geq 10$.
- Appendix F.2: the performance of NCE is less sensitive to the selection of intra solvers.
- Appendix F.3: the performance of NCE is less sensitive to the selection of swapping tours.
- Appendix F.4: the performance of NCE converges when the perturbation parameter $p \geq 5$.

## 5 RELATED WORKS

**Supervised learning (SL) approach to solve VRPs**   SL approaches (Joshi et al., 2019; Vinyals et al., 2015; Xin et al., 2021b; Li et al., 2021; 2018) utilize the supervision from the VRP solvers as the training labels. (Vinyals et al., 2015; Joshi et al., 2019) imitates TSP solvers using PointerNet and graph convolution network (GCN), respectively. (Joshi et al., 2019) trains a GCN to predict the edge occurrence probabilities in TSP solutions. Even though SL often offer a faster solving speed than existing solvers, their use is limited to the problems where the solvers are available. Such property limits the use of SL from general and realistic VRPs.

**Reinforcement learning (RL) approach to solve VRPs**   RL approaches (Bello et al., 2016; Khalil et al., 2017; Nazari et al., 2018; Kool et al., 2018; Kwon et al., 2020; Park et al., 2021; Cao et al., 2021; Guo et al., 2019; Wu et al., 2019; 2021; Falkner & Schmidt-Thieme, 2020; Chen & Tian, 2019) exhibit promising performances that are comparable to existing solvers as they learn solvers from the problem-solving simulations. (Bello et al., 2016; Nazari et al., 2018; Kool et al., 2018; Guo et al., 2019) utilize an encoder-decoder structure to generate routing schedules sequentially, while (Park et al., 2021; Khalil et al., 2017) use graph-based embedding to determine the next assignment action. However, RL approaches often require the problem-specific Markov decision process and network design. NCE does not require the simulation of the entire problem-solving. Instead, it only requires the computation of the swapping operation (i.e., the results of CE). This property allows NCE to be trained easily to solve various routing problems with one scheme.

**Neural network-based (meta) heuristic approach**   Combining machine learning (ML) components with existing (meta) heuristics shows strong empirical performances when solving VRPs (Hottung & Tierney, 2019; Xin et al., 2021b; Li et al., 2021; Lu et al., 2019; da Costa et al., 2021; Kool et al., 2021). They often employ ML to learn to solve NP-hard sub-problems of VRPs, which are difficult. For example, L2D (Li et al., 2021) learns to predict the objective value of CVRP, NLNS (Hottung & Tierney, 2019) learns a TSP solver when solving VRPs and DPDP (Kool et al., 2021) learns to boost the dynamic programming algorithms. To learn such solvers, these methods apply SL or RL. Instead, NCE learns the fundamental operator of meta-heuristics rather than predict or generate a solution. Hence, NCE that is trained on FMDVRP generalizes well to the special cases of FMDVRP. Furthermore, the training data for NCE can be prepared effortlessly.

## 6 CONCLUSION

We propose Neuro CROSS exchange (NCE), a neural network-enhanced CE operator, to learn a fundamental and universal operator that can be used to solve the various types of min-max VRPs. We validated that NCE can solve various min-max VRPs without retraining for each specific problem, exhibiting strong empirical performances. Although NCE addresses more realistic VRPs (i.e., min-max FMDVRP) than existing NCO solvers, NCE does not yet consider complex constraints such as pickup and delivery, and time windows. Our future research will focus on solving more complex VRPs by considering such various constraints during the NCE operation.

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

# Neuro CROSS exchange
# *Supplementary Material*

## A    MILP FORMULATIONS FOR MIN-MAX ROUTING PROBLEMS

This section provides the mixed integer linear programming (MILP) formulations of mTSP, MD-VRP, and FMDVRP.

### A.1    MTSP

mTSP is a multi-vehicle extension of the traveling salesman problem (TSP). mTSP comprises of a set of the nodes (i.e., cities) and depot $V$, a set of vehicles $K$, and a set of depot $S$. We define $d_{ij}$ as the cost (or travel time) between node $i$ and $j$, and the decision variable $x_{ijk}$ denotes whether the edge between node $i$ and $j$ are taken by vehicle $k$. Following the convention, we consider mTSP with $|S| = 1$. The MILP formulation of mTSP is given as follows:

$$\text{minimize} \quad Q \tag{A.1}$$

$$\text{subject to.} \quad \sum_{i \in V} \sum_{j \in V} d_{ij} x_{ijk} \leq Q, \qquad \forall k \in K : i \neq j, \tag{A.2}$$

$$\sum_{j \in V i \neq j} x_{ijk} = 1, \qquad \forall k \in K, \forall i \in S, \tag{A.3}$$

$$\sum_{i \in V j \neq i} \sum_{k \in T} x_{ijk} = 1, \qquad \forall j \in V \setminus S \tag{A.4}$$

$$\sum_{i \in V i \neq j} x_{ijk} - \sum_{h \in V h \neq j} x_{jhk} = 0, \qquad \forall j \in V \setminus S \tag{A.5}$$

$$u_{ik} - u_{jk} + |V| x_{ijk} \leq |V| - 1, \qquad \forall k \in K, j \in V \setminus S : i \neq j, \tag{A.6}$$

$$0 \leq u_{ik} \leq |V| - 1, \qquad \forall k \in K, i \in V \setminus S \tag{A.7}$$

$$x_{ijk} \in \{0, 1\}, \qquad \forall k \in K, \forall i, j \in V, \tag{A.8}$$

$$u_{ik} \in \mathbb{Z}, \qquad \forall k \in K, i \in V \tag{A.9}$$

where $Q$ denotes the longest traveling distance among multiple vehicles. (i.e., makespan), Eq. (A.3) indicates the vehicles start at the depot, Eq. (A.4) indicates that all cities are visited, Eq. (A.5) indicates the balance equation for all cities, Eq. (A.6) and Eq. (A.7) indicate the sub-tour eliminations.

## A.2 MDVRP

Multi-depot VRP is a multi-depot extension of mTSP (Appendix A.1), where each vehicle starts from its own designated depot and returns to the depot. We extend the MILP formulation of mTSP to define the MILP formulation of MDVRP. On top of the mTSP formulation, we define $K_i$, which indicates the set of vehicles assigned to the depot $i$.

$$
\begin{align}
\text{minimize} \quad & Q \tag{A.10}\\
\text{subject to.} \quad & \sum_{i \in V} \sum_{j \in V} d_{ij} x_{ijk} \leq Q, & \forall k \in K : i \neq j, \tag{A.11}\\
& \sum_{j \in V \, j \neq i} \sum_{k \in T} x_{ijk} = 1, & \forall i \in V \setminus S \tag{A.12}\\
& \sum_{i \in V \, j \neq i} \sum_{k \in T} x_{ijk} = 1, & \forall j \in V \setminus S \tag{A.13}\\
& \sum_{i \in V} x_{ijk} - \sum_{h \in V} x_{jhk} = 0, & \forall j \in V \setminus S, \forall k \in K \tag{A.14}\\
& u_{ik} - u_{jk} + |V| x_{ijk} \leq |V| - 1, & \forall k \in K, j \in V \setminus S : i \neq j, \tag{A.15}\\
& 0 \leq u_{ik} \leq |V| - 1, & \forall k \in K, i \in V \setminus S \tag{A.16}\\
& x_{ijk} \in \{0, 1\}, & \forall k \in K, \forall i, j \in V, \tag{A.17}\\
& u_{ik} \in \mathbb{Z}, & \forall k \in K, i \in V \tag{A.18}\\
& \sum_{j \in V \setminus S} x_{ijk} \leq 1, & \forall k \in K_i, \forall i \in S \tag{A.19}\\
& \sum_{i \in V \setminus S} x_{ijk} \leq 1, & \forall k \in K_j, \forall j \in S \tag{A.20}
\end{align}
$$

where Eq. (A.19) and Eq. (A.20) indicate that each vehicle starts and returns its own depot at most once.

## A.3 FMDVRP

Flexible MDVRP (FMDVRP) is an extension of MDVRP, allowing the vehicle to return to any depot. We extend the MDVRP formulation (Appendix A.2) to define the FMDVRP formulation. To account for the flexibility of the returning depot, we introduce a dummy node for all depots; a depot is modeled with a start and return depot. We define $S_1$ and $S_2$ as the set of start and return depots and $s_k$ as the start node of the vehicle $k$.

$$\text{minimize} \quad Q \tag{A.21}$$

$$\text{subject to.} \quad \sum_{i \in V} \sum_{j \in V} d_{ij} x_{ijk} \leq Q, \qquad \forall k \in K : i \neq j, \tag{A.22}$$

$$\sum_{j \in V j \neq i} \sum_{k \in T} x_{ijk} = 1, \qquad \forall i \in V \setminus S \tag{A.23}$$

$$\sum_{i \in V j \neq i} \sum_{k \in T} x_{ijk} = 1, \qquad \forall j \in V \setminus S \tag{A.24}$$

$$\sum_{i \in V} x_{ijk} - \sum_{h \in V} x_{jhk} = 0, \qquad \forall j \in V \setminus S, \forall k \in K \tag{A.25}$$

$$u_{ik} - u_{jk} + |V| x_{ijk} \leq |V| - 1, \qquad \forall k \in K, j \in V \setminus S : i \neq j, \tag{A.26}$$

$$0 \leq u_{ik} \leq |V| - 1, \qquad \forall k \in K, i \in V \setminus S \tag{A.27}$$

$$x_{ijk} \in \{0, 1\}, \qquad \forall k \in K, \forall i, j \in V, \tag{A.28}$$

$$u_{ik} \in \mathbb{Z}, \qquad \forall k \in K, i \in V \tag{A.29}$$

$$\sum_{j \in V \setminus S} x_{s_k jk} = 1, \qquad \forall k \in K \tag{A.30}$$

$$\sum_{j \in V \setminus S} x_{ijk} = 0, \qquad \forall k \in K, \forall i \in S \setminus s_k \tag{A.31}$$

$$\sum_{j \in V \setminus S} x_{ijk} \leq 1, \qquad \forall k \in K_i, \forall i \in S1 \tag{A.32}$$

$$\sum_{i \in V \setminus S} x_{ijk} \leq 1, \qquad \forall k \in K_j, \forall j \in S2 \tag{A.33}$$

$$\sum_{j \in V \setminus S} x_{ijk} = 0, \qquad \forall k \in K, \forall i \in S2 \tag{A.34}$$

$$\sum_{j \in V \setminus S} x_{ijk} = 0, \qquad \forall k \in K, \forall i \in S1 \tag{A.35}$$

$$\sum_{i \in S1} \sum_{j \in V \setminus S} x_{ijk} = \sum_{i \in V \setminus S} \sum_{j \in S2} x_{ijk}, \qquad \forall k \in K \tag{A.36}$$

where Eqs. (A.30) and (A.31) indicate that each vehicle starts at its own depot. Eqs. (A.32) to (A.35) indicate the start and return depots constraints. Eq. (A.36) indicates the balance equation of the start and return depots.

# B MIN-MAX MDVRP RESULTS

In this section, we provide the experiment results of MDVRP. We apply NCE with the $f_\theta$ trained on FMDVRP instances to solve MDVRP. For each $(N_c, N_d, N_v)$ pair, we measure the average makespan of 100 instances. We provide the MDVRP results in Tables A.1 to A.3. Similar to the FMDVRP experiments, NCE shows leading performance while being faster than the baselines. From the results, we can conclude that the learned $f_\theta$ is transferable to the different problem sets. This phenomenon is rare in many ML-based approaches. It again highlights the effectiveness of learning fundamental operators (i.e., learn what should be cross exchanged) when solving the VRP families.

Table A.1: **MDVRP results** (small-sized instances)

| $N_c,N_d$ ($\downarrow$) | $N_v(\rightarrow)$ Method | Cost | 2 Gap(%) | Time(sec.) | Cost | 3 Gap(%) | Time(sec.) |
|---|---|---|---|---|---|---|---|
| (7,2) | CPLEX | **1.626** | 0.00 | 0.32 | **1.417** | 0.00 | 0.54 |
| | OR-tools | 1.704 | 4.80 | 0.01 | 1.433 | 1.13 | 0.01 |
| | CE | **1.626** | 0.00 | 0.05 | 1.418 | 0.01 | 0.04 |
| | NCE | **1.626** | 0.00 | 0.13 | 1.418 | 0.01 | 0.16 |
| | $N_v(\rightarrow)$ Method | Cost | 2 Gap(%) | Time(sec.) | Cost | 3 Gap(%) | Time(sec.) |
| (10,2) | CPLEX | **1.829** | 0.00 | 7.90 | **1.554** | 0.00 | 33.17 |
| | OR-tools | 1.926 | 5.30 | 0.02 | 1.590 | 2.32 | 0.02 |
| | CE | **1.829** | 0.00 | 0.09 | 1.558 | 0.03 | 0.08 |
| | NCE | **1.829** | 0.00 | 0.17 | 1.555 | 0.01 | 0.20 |

Table A.2: **MDVRP results** (medium-sized instances)

| $N_c, N_d$ ($\downarrow$) | $N_v(\rightarrow)$ Method | Cost | 3 Gap(%) | Time(sec.) | Cost | 5 Gap(%) | Time(sec.) | Cost | 7 Gap(%) | Time(sec.) |
|---|---|---|---|---|---|---|---|---|---|---|
| (50,6) | OR-tools | 2.64 | 17.33 | 2.24 | 1.68 | 9.80 | 2.94 | 1.36 | 6.25 | 2.75 |
| | CE | 2.25 | 0.00 | 23.45 | 1.53 | 0.00 | 10.40 | 1.28 | 0.00 | 6.85 |
| | NCE | **2.25** | 0.00 | 2.08 | **1.53** | 0.00 | 2.63 | **1.28** | 0.00 | 2.93 |
| | $N_v(\rightarrow)$ Method | Cost | 5 Gap(%) | Time(sec.) | Cost | 7 Gap(%) | Time(sec.) | Cost | 10 Gap(%) | Time(sec.) |
| (100,8) | OR-tools | 2.17 | 17.30 | 33.08 | 1.60 | 11.89 | 36.45 | 1.29 | 9.32 | 37.54 |
| | CE | **1.85** | 0.00 | 259.82 | 1.43 | 0.00 | 140.63 | 1.18 | 0.00 | 86.27 |
| | NCE | 1.86 | 0.54 | 11.61 | **1.43** | 0.00 | 11.96 | **1.18** | 0.00 | 15.70 |

Table A.3: **MDVRP results** (large-sized instances)

| $N_c, N_d, N_v(\rightarrow)$ Method | Cost | 200,10,10 Gap(%) | Time(sec.) | Cost | 400,20,20 Gap(%) | Time(sec.) | Cost | 600,30,30 Gap(%) | Time(sec.) |
|---|---|---|---|---|---|---|---|---|---|
| OR-tools | 1.49 | 9.56 | 452.08 | 1.13 | 13.00 | 5179 | 1.01 | 24.69 | 22411 |
| Greedy | 2.99 | 119.85 | 0.20 | 2.77 | 177.00 | 0.93 | 3.20 | 295.06 | 2.69 |
| Greedy + TSP | 2.78 | 104.41 | 0.37 | 2.58 | 158.00 | 1.48 | 3.03 | 274.07 | 3.24 |
| NCE | **1.36** | 0.00 | 77.73 | **1.00** | 0.00 | 553.83 | **0.81** | 0.00 | 3173.5 |

## C  MIN-MAX MTSP RESULTS

In this section, we provide the additional experiment results of mTSP. We further apply NCE to solve mTSPLib (mTSPLib), which comprises of mTSP instances from real cities, and large scale problem. As reported in Table A.4 and Table A.5 NCE achieves the best results as compared to the baselines.

Table A.4: **mTSPLib results**: CPLEX results with $*$ are optimal solutions. Otherwise, the known-best upper bound of CPLEX results are reported. The results of other baselines are taken from Park et al. (2021).

| $N_c(\rightarrow)$ | Eil51 | | | | Berlin52 | | | | Eil76 | | | | Rat99 | | | | |
|---|---|---|---|---|---|---|---|---|---|---|---|---|---|---|---|---|---|
| $N_v(\rightarrow)$ | 2 | 3 | 5 | 7 | 2 | 3 | 5 | 7 | 2 | 3 | 5 | 7 | 2 | 3 | 5 | 7 | Gap |
| CPLEX | 222.7* | 159.6 | 124.0 | 112.1 | 4110 | 3244 | 2441 | 2441 | 280.9* | 197.3 | 150.3 | 139.6 | 728.8 | 587.2 | 469.3 | 443.9 | 1.00 |
| LKH-3 | **222.7** | **159.6** | 124.0 | 112.1 | 4110 | 3244 | **2441** | 2441 | **280.9** | **197.3** | 150.3 | 139.6 | 728.8 | 587.2 | 469.3 | 443.9 | 1.00 |
| OR-Tools | 243.0 | 170.1 | 127.5 | 112.1 | 4665 | 3311 | 2482 | 2441 | 318.0 | 212.4 | 143.4 | 128.3 | 762.2 | 552.1 | 473.7 | 442.5 | 1.03 |
| ScheduleNet | 263.9 | 200.5 | 131.7 | 116.9 | 4826 | 3644 | 2758 | 2515 | 330.2 | 228.8 | 163.9 | 144.4 | 843.8 | 691.8 | 524.3 | 480.8 | 1.13 |
| ScheduleNet (s.64) | 239.3 | 173.5 | 125.8 | 112.2 | 4592 | 3276 | 2517 | 2441 | 317.7 | 220.8 | 153.8 | 131.7 | 781.2 | 627.1 | 502.3 | 464.4 | 1.05 |
| DAN | 274.2 | 178.9 | 158.6 | 118.1 | 5226 | 4278 | 2759 | 2697 | 361.1 | 251.5 | 170.9 | 148.5 | 930.8 | 674.1 | 504.0 | 466.4 | 1.18 |
| DAN (s.64) | 252.9 | 178.9 | 128.2 | 114.3 | 5098 | 3456 | 2677 | 2495 | 336.7 | 228.1 | 157.9 | 134.5 | 966.5 | 697.7 | 495.6 | 462.0 | 1.11 |
| NCE | 235.0 | 170.3 | 121.6 | **112.1** | **4110** | 3274 | 2660 | **2441** | 285.5 | 211.0 | **144.6** | 127.6 | 695.8 | **527.8** | **458.6** | **441.6** | 1.00 |
| NCE-mTSP | 226.1 | 166.3 | **119.9** | 112.1 | 4128 | **3191** | 2474 | **2441** | 282.1 | 197.5 | 147.2 | **127.6** | **666.0** | 533.2 | 462.2 | 443.9 | **0.98** |

Table A.5: **mTSP results** (large-sized instances)

| $N_c(\rightarrow)$ Method | mTSP300 | | | mTSP500 | | | mTSP750 | | |
|---|---|---|---|---|---|---|---|---|---|
| | Cost | Gap(%) | Time(sec.) | Cost | Gap(%) | Time(sec.) | Cost | Gap(%) | Time(sec.) |
| LKH-3 | 2.12 | 0.00 | 1620 | 2.09 | 0.00 | 2981 | 1.99 | 0.00 | 6303 |
| OR-tools | 2.90 | 36.79 | 3605 | 7.63 | 265.07 | 9877 | 10.91 | 448.24 | 47225 |
| ScheduleNet | 2.25 | 6.13 | 689 | 2.17 | 3.83 | 3788 | 2.12 | 6.53 | 12199 |
| Greedy | 2.82 | 33.02 | 1.07 | 2.85 | 36.36 | 3.78 | 2.76 | 38.69 | 4.56 |
| Greedy + TSP | 2.80 | 32.08 | 1.39 | 2.84 | 35.89 | 4.48 | 2.75 | 38.19 | 5.53 |
| NCE | **2.12** | 0.00 | 301 | **2.09** | 0.00 | 583 | **1.99** | 0.00 | 580 |

# D    MIN-MAX CVRP RESULTS

We also provide min-max CVRP results for CVRPlib. We include two local search heuristic baselines N2 (Pham et al., 2017) and NMCF (Van Nguyen et al., 2017). We evaluate each instance 20 times and average it. As shown in Table A.6, NCE outperforms other Operation Research baselines in terms of computation speed and cost.

Table A.6: **CVRPlib result:** Results are taken from Van Nguyen et al. (2017)

| Method | N2 | | | NMCF | | | NCE (s.10) | | |
|---|---|---|---|---|---|---|---|---|---|
| VRP instance | Cost | Gap(%) | Time(sec.) | Cost | Gap(%) | Time(sec.) | Cost | Gap(%) | Time(sec.) |
| E-n22-k4 | **110.00** | -0.63 | 4.98 | **110.00** | -0.63 | 3.62 | 110.70 | 0.00 | 2.51 |
| E-n23-k3 | 242.10 | 0.04 | 20.50 | 243.45 | 0.60 | 27.35 | **242.00** | 0.00 | 14.09 |
| E-n30-k3 | 192.95 | 0.63 | 53.72 | 192.45 | 0.37 | 10.85 | **191.75** | 0.00 | 10.45 |
| E-n33-k4 | 244.15 | 0.00 | 85.87 | **244.00** | -0.06 | 11.07 | 244.15 | 0.00 | 6.43 |
| E-n51-k5 | 118.70 | 5.42 | 119.50 | 113.50 | 0.80 | 67.09 | **112.60** | 0.00 | 14.40 |
| E-n76-k7 | 119.35 | 12.44 | 115.85 | 107.35 | 1.13 | 120.49 | **106.15** | 0.00 | 34.94 |
| E-n76-k8 | 118.85 | 18.67 | 140.44 | **100.05** | -0.10 | 94.57 | 100.15 | 0.00 | 31.27 |
| E-n76-k10 | 117.40 | 23.71 | 149.21 | 99.55 | 4.90 | 31.69 | **94.90** | 0.00 | 34.94 |
| E-n76-k14 | 111.10 | 27.12 | 146.64 | 93.60 | 7.09 | 27.22 | **87.40** | 0.00 | 42.13 |
| E-n101-k8 | 141.65 | 26.42 | 140.07 | 117.05 | 4.46 | 69.01 | **112.05** | 0.00 | 71.96 |
| E-n101-k14 | 113.25 | 13.19 | 161.05 | 101.50 | 1.45 | 68.04 | **100.05** | 0.00 | 83.74 |

# E MIN-SUM CVRP RESULTS

In this section, we provide the results for min-sum CVRP benchmark instances. To solve min-sum CVRP, we trained NCE to predict min-sum cost decrement with $C(\tau_1, \tau_2)$ defined as $l(\tau_1) + l(\tau_2)$. As shown in Table A.7, NCE is on par with or outperforms other neural baselines. However, NCE is not efficient in terms of computation speed. This is because we initially design the CE operation to solve the min-max VRPs *not* min-sum VRPs. According to the experimental results, when CE operation is conducted to reduce the min-sum cost of two tours, it typically requires larger iterations to improve the solution to near optimum. This results indicates that other (local) search operator is required to efficiently solve the min-sum problems.

Table A.7: **CVRP benchmark results:** Best in **bold**; Second underline, $(s.n)$ indicates the best results of $n$ sampling, $(i.n)$ indicates the best results after $n$ improvement steps, and † indacates that the computation times of the neural baselines are measured with GPU. The run times of Lu et al. (2019) and Kwon et al. (2020) are taken from Kwon et al. (2020). The run times of the other neural baselines are taken from Kim et al. (2021).

| Method | CVRP20 | | | CVRP50 | | | CVRP100 | | |
|---|---|---|---|---|---|---|---|---|---|
| | Cost | Gap(%) | Time(sec.) | Cost | Gap(%) | Time(sec.) | Cost | Gap(%) | Time(sec.) |
| LKH-3 | 6.14 | 0.00 | 0.72 | **10.38** | 0.00 | 2.52 | **15.65** | 0.00 | 4.68 |
| OR-Tools | 6.43 | 4.72 | 0.01 | 11.31 | 8.17 | 0.05 | 17.16 | 10.29 | 0.23 |
| RL†$(s.10)$ (Nazari et al., 2018) | 6.40 | 4.23 | 0.16 | 11.15 | 7.46 | 0.23 | 16.96 | 8.39 | 0.45 |
| AM†$(s.1280)$ (Kool et al., 2018) | 6.25 | 1.79 | 0.05 | 10.62 | 2.40 | 0.14 | 16.23 | 3.72 | 0.34 |
| MDAM†$(s.50)$ (Xin et al., 2021a) | 6.14 | 0.00 | 0.03 | 10.48 | 0.96 | 0.09 | 15.99 | 2.17 | 0.32 |
| POMO†$(s.8)$ (Kwon et al., 2020) | 6.14 | 0.00 | 0.01 | 10.42 | 0.35 | 0.01 | 15.73 | 0.43 | 0.01 |
| NLNS†$(i.1280)$ (Hottung & Tierney, 2019) | 6.19 | 0.81 | 1.00 | 10.54 | 1.54 | 1.63 | 16.00 | 2.24 | 2.18 |
| AM + LCP†$(s.1280)$ (Kim et al., 2021) | 6.16 | 0.33 | 0.09 | 10.54 | 1.54 | 0.20 | 16.03 | 2.43 | 0.45 |
| Learn2OPT†$(i.2000)$ (da Costa et al., 2021) | 6.16 | 0.37 | 1.00 | 10.54 | 2.66 | 1.44 | 16.72 | 6.40 | 7.20 |
| LIH†$(i.5000)$ (Wu et al., 2021) | **6.12** | -0.33 | 0.72 | 10.45 | 0.67 | 1.44 | 16.03 | 2.43 | 1.80 |
| NeuroRewriter† (Chen & Tian, 2019) | 6.16 | 0.48 | 0.13 | 10.51 | 1.25 | 0.21 | 16.10 | 2.88 | 0.40 |
| NCE | 6.21 | 1.14 | 0.73 | 10.68 | 2.89 | 3.23 | 16.29 | 4.09 | 13.67 |
| CE | 6.21 | 1.14 | 0.77 | 10.69 | 2.99 | 11.11 | 16.28 | 4.03 | 79.12 |
| NCE (s.10, p.20) | 6.13 | -0.16 | 3.99 | 10.41 | 0.29 | 20.17 | 15.81 | 1.02 | 90.12 |

## F ABLATION STUDY

In this section, we provide the results of the ablation studies.

### F.1 CANDIDATE SET

NCE constructed a search candidate set. To mitigate the prediction error of $f_\theta(\cdot)$ in finding the argmax of $(a_1, a_2, b_1, b_2)$, NCE search the top $K$ pairs of $(a_1, a_2)$ that have the largest $y^*$ out of all $(a_1, a_2)$ choices. We measure how the performance changes whenever the size of the candidate set $K$ changes. As shown in Table A.8, as the size of $K$ increases, the performance tends to increase slightly. When $K \geq 10$, the performance of NCE almost converges. Thus, we choose $K = 10$ as the default hyperparameter of NCE.

Table A.8: **Effect of number of candidate**

| $K$ | 1 | | 2 | | 3 | | 5 | | 7 | | 10 | | 20 | | 30 | |
|---|---|---|---|---|---|---|---|---|---|---|---|---|---|---|---|---|
| $N_c,N_d,N_v$ | cost | time | cost | time | cost | time | cost | time | cost | time | cost | time | cost | time | cost | time |
| (30,3,2) | 2.47 | 0.26 | 2.44 | 0.30 | 2.44 | 0.34 | 2.43 | 0.38 | 2.43 | 0.43 | 2.43 | 0.48 | 2.43 | 0.63 | 2.43 | 0.81 |
| (30,3,3) | 1.87 | 0.27 | 1.85 | 0.31 | 1.84 | 0.35 | 1.84 | 0.41 | 1.83 | 0.48 | 1.83 | 0.55 | 1.83 | 0.79 | 1.83 | 1.04 |
| (30,3,5) | 1.50 | 0.50 | 1.47 | 0.61 | 1.47 | 0.66 | 1.46 | 0.71 | 1.46 | 0.83 | 1.46 | 0.91 | 1.47 | 1.27 | 1.46 | 1.54 |
| (50,3,3) | 2.23 | 0.58 | 2.20 | 0.80 | 2.19 | 0.93 | 2.18 | 1.13 | 2.19 | 1.26 | 2.18 | 1.51 | 2.19 | 2.17 | 2.18 | 2.70 |
| (50,3,5) | 1.67 | 0.86 | 1.63 | 1.12 | 1.62 | 1.34 | 1.61 | 1.56 | 1.61 | 1.81 | 1.61 | 2.17 | 1.61 | 3.14 | 1.61 | 4.07 |
| (50,3,7) | 1.49 | 1.05 | 1.47 | 1.31 | 1.47 | 1.59 | 1.46 | 1.93 | 1.46 | 2.18 | 1.46 | 2.59 | 1.46 | 3.79 | 1.46 | 4.98 |

### F.2 INTRA-SOLVER

NCE repeatedly applies the inter- and intra-operation. In this view, the choice of the intra-operation may affect the performance of NCE. In this subsection, we measured the performance of NCE according to intra-operation. We compare the results of NCE that uses Elkai, OR-tools, and 2-opt as the intra-operator. To solve TSP – the task intra-operator has to solve –, Elkai, OR-tools, and 2-opt show the best, second best, and third best performances. As shown in Table A.9, the performances of NCE are almost identical to the selection of an intra-operator. We validate that the effect of the intra-operation choice is negligible to the performance.

Table A.9: **Effect of Intra TSP solver**

| $N_c,N_d,N_v$ | (30,3,2) | | (30,3,3) | | (30,3,5) | | (50,3,3) | | (50,3,5) | | (50,3,7) | |
|---|---|---|---|---|---|---|---|---|---|---|---|---|
| Intra solver | cost | time | cost | time | cost | time | cost | time | cost | time | cost | time |
| 2-opt | 2.46 | 0.23 | 1.83 | 0.33 | 1.47 | 0.53 | 2.22 | 0.72 | 1.62 | 1.24 | 1.46 | 1.58 |
| OR-tools | 2.44 | 1.04 | 1.83 | 1.08 | 1.47 | 1.06 | 2.20 | 3.31 | 1.61 | 2.72 | 1.46 | 2.72 |
| Elkai | 2.43 | 0.41 | 1.83 | 0.55 | 1.46 | 0.69 | 2.18 | 1.55 | 1.61 | 2.17 | 1.46 | 2.13 |

### F.3 SELECTING TWO VEHICLES

NCE chooses two tours for improvement during the iterative process. To understand the effect of the tour selection strategy, we measure the performance of NCE according to tour selection. We compare NCE results in a max-min selection case and a random selection case (i.e., pick two tours randomly). As shown in Table A.10, the performances of NCE are almost identical to the tour selection strategy. Therefore, we validate that the effect of the tour selection strategy is negligible.

Table A.10: **Effect of selecting two vehicles**

| $N_c,N_d,N_v$ | (30,3,2) | | (30,3,3) | | (30,3,5) | | (50,3,3) | | (50,3,5) | | (50,3,7) | |
|---|---|---|---|---|---|---|---|---|---|---|---|---|
| | cost | time | cost | time | cost | time | cost | time | cost | time | cost | time |
| Random | 2.43 | 0.42 | 1.84 | 0.61 | 1.48 | 0.94 | 2.18 | 1.43 | 1.62 | 2.39 | 1.47 | 2.62 |
| Max-Min | 2.43 | 0.41 | 1.83 | 0.55 | 1.46 | 0.69 | 2.18 | 1.55 | 1.61 | 2.17 | 1.46 | 2.13 |

### F.4 PERTURBATION

NCE employs perturbation to increase performance. Perturbation is a commonly used strategy for enhancing the performance of meta-heuristics (Polat et al., 2015). It is done by randomly *perturbing* the solution and solving the problem with the perturbed solutions. This technique is beneficial to escape from the local optima. As described in Algorithm 1, when falling into the local optima, NCE randomly selects two tours and performs a random exchange. We compare the performance of NCE with different perturbations. As shown in Table A.11, the performance of NCE increases and converges as the number of perturbations $p$ increases. When $p = 5$, the performance of NCE converges. Thus, we choose $p = 5$ as the default hyperparameter of NCE.

Table A.11: **Effect of perturbation**

| $P$ | 0 | | 1 | | 2 | | 3 | | 5 | | 7 | | 10 | | 20 | |
|---|---|---|---|---|---|---|---|---|---|---|---|---|---|---|---|---|
| $N_c,N_d,N_v$ | cost | time | cost | time | cost | time | cost | time | cost | time | cost | time | cost | time | cost | time |
| (30,3,2) | 2.50 | 0.12 | 2.48 | 0.17 | 2.46 | 0.22 | 2.44 | 0.27 | 2.43 | 0.38 | 2.43 | 0.49 | 2.42 | 0.69 | 2.41 | 1.34 |
| (30,3,3) | 1.89 | 0.16 | 1.86 | 0.22 | 1.84 | 0.30 | 1.84 | 0.35 | 1.83 | 0.55 | 1.82 | 0.61 | 1.81 | 0.81 | 1.81 | 1.42 |
| (30,3,5) | 1.49 | 0.29 | 1.48 | 0.33 | 1.48 | 0.43 | 1.47 | 0.50 | 1.47 | 0.67 | 1.46 | 0.85 | 1.46 | 1.25 | 1.46 | 2.28 |
| (50,3,3) | 2.26 | 0.31 | 2.24 | 0.49 | 2.22 | 0.65 | 2.19 | 0.81 | 2.18 | 1.28 | 2.17 | 1.83 | 2.16 | 2.61 | 2.14 | 4.83 |
| (50,3,5) | 1.66 | 0.52 | 1.64 | 0.77 | 1.63 | 0.94 | 1.62 | 1.24 | 1.61 | 1.97 | 1.61 | 2.59 | 1.60 | 3.61 | 1.59 | 6.53 |
| (50,3,7) | 1.48 | 0.85 | 1.48 | 1.04 | 1.47 | 1.43 | 1.47 | 2.02 | 1.46 | 2.65 | 1.46 | 2.95 | 1.46 | 3.77 | 1.45 | 6.44 |

## G  TRAINING DETAIL

**Dataset preparation.**  To train the cost-decrement prediction model $f_\theta(\cdot)$, we generate 50,000 random FMDVRP instances. The random instance is generated by first sampling the number of customer $N_c$ and depots $N_d$ from $\mathcal{U}(10, 100)$ and $\mathcal{U}(2, 9)$ and $N_v = 2$ respectively, and then sampling the 2D coordinates of the cities from $\mathcal{U}(0, 1)$. As we set $N_v = 2$, we generate two tours by applying the initial solution construction heuristics explained in Section 3.1. From $\tau_1, \tau_2$, we compute the best true cost-decrements of all feasible $(a_1, a_2)$ to prepare the training dataset. We generated 47,856,986 training samples from the 50,000 instances.

**Hyperparameters.**  $f_\theta(\cdot)$ is parametrized via the GNN which employs five layers of the attentive embedding layer. We employ 4 layered MLPs to parameterize $\phi_e, \phi_w, \phi_n$ and $\phi_c$, whose hidden dimensions and activation units are 64 and Mish (Misra, 2019). $f_\theta(\cdot)$ is trained to minimize the Huber loss for three epochs via AdamW (Loshchilov & Hutter, 2017) whose learning rate is fixed as $5 \times 10^{-4}$.

**Computing resources.**  We run all experiments on the server equipped with AMD Threadripper 2990WX CPU and Nvidia RTX 3090 GPU. We use a single CPU core to evaluate all algorithms.

## H    EVALUATION OF THE COST DECREMENT MODEL

In this section, we evaluate the prediction accuracy of $f_\theta(\cdot)$. To evaluate $f_\theta(\cdot)$, we randomly generate 1,000 FMDVRP instances by sampling $N_C \sim \mathcal{U}(10, 100)$ and $N_D \sim \mathcal{U}(2, 9)$, and $(x, y) \sim \mathcal{U}(0, 1)^2$. From the instances, we measure the ratio of existence of the argmax $(a_1, a_2)$ pair in the search candidate set whose size is $K$. As shown in Table A.12, when $K \geq 10$, NCE can find the argmax pair with at least $0.9$ probability. We also provide the results of the cost-decrement predictions and its corresponding cost. As shown in Fig. A.1, $f_\theta(\cdot)$ predicts the general tendency well.

Table A.12: $f_\theta(\cdot)$ **prediction performance test**

| $K$ | 1 | 3 | 5 | 10 | 20 |
|---|---|---|---|---|---|
| argmax ratio (%) | 42.9 | 71.3 | 78.6 | 90.9 | 97.4 |

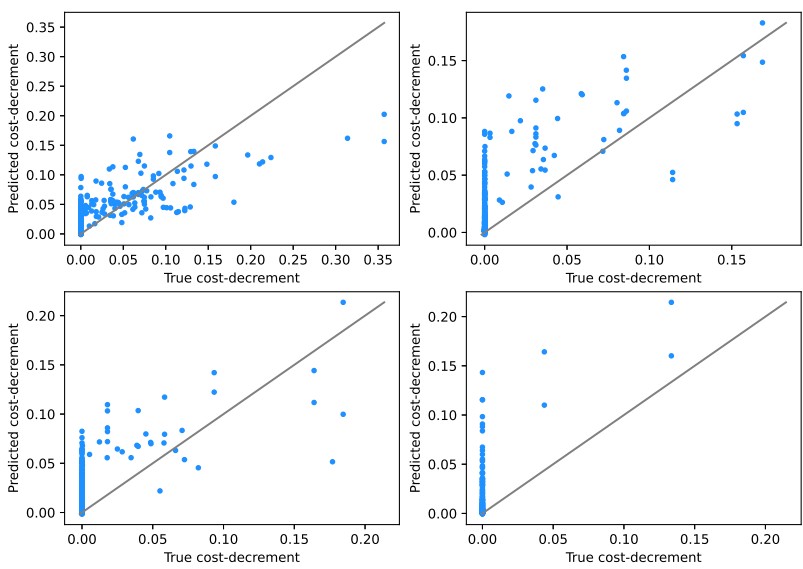

Figure A.1: Predicted cost-decrements vs. true cost-decrements

# I    COMPARISON WITH FULL SEARCH

To verify whether NCE successfully amotrizes CE, we evaluate CE and NCE($K$=10, $p$=0) on FMDVRP. For the testing instances, we randomly generate 100 instances for each $N_c \in \{20, 30, 40, 50, 60, 70, 80, 90, 100\}$ with the fixed $N_d = 3$ and $N_v = 3$. As shown in Table A.13, NCE shows nearly identical performances with CE, but with significantly faster computation speed than CE as shown in Fig. A.2. We also test the mTSP and CVRP cases as shown in Table A.14.

Table A.13: **FMDVRP performance comparison of CE and NCE($K$=10, $p$=0)**

| $N_d$ ,$N_v$ | | | | | (3,3) | | | | |
|---|---|---|---|---|---|---|---|---|---|
| $N_C$ | 20 | 30 | 40 | 50 | 60 | 70 | 80 | 90 | 100 |
| CE | 1.651 | 1.893 | 2.088 | 2.257 | 2.384 | 2.531 | 2.695 | 2.811 | 2.929 |
| NCE | 1.651 | 1.891 | 2.088 | 2.262 | 2.390 | 2.530 | 2.697 | 2.806 | 2.934 |

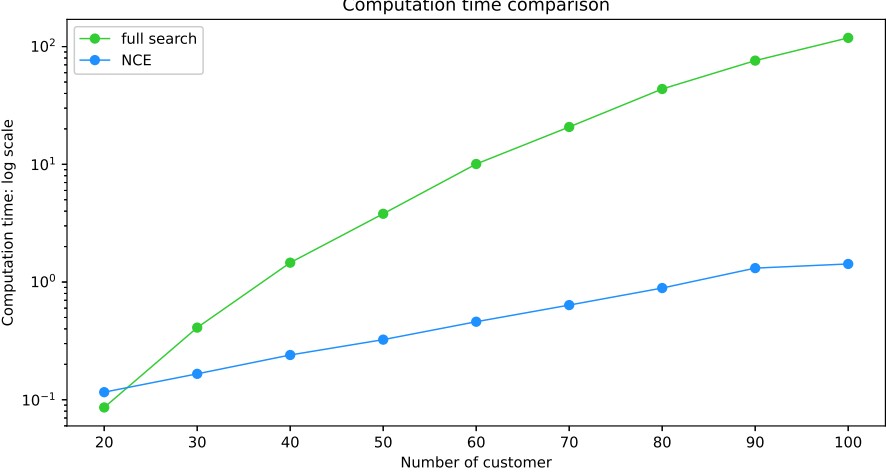

Figure A.2: Computation speed comparison

Table A.14: **mTSP, CVRP performance comparison of CE and NCE**

| VRP instance | mTSP50,$N_v$=5 | | mTSP100,$N_v$=10 | | mTSP200,$N_v$=15 | | CVRP20 | | CVRP50 | | CVRP100 | |
|---|---|---|---|---|---|---|---|---|---|---|---|---|
| | cost | time | cost | time | cost | time | cost | time | cost | time | cost | time |
| CE | 2.02 | 11.35 | 1.97 | 87.28 | 2.00 | 414.9 | 6.21 | 0.96 | 10.91 | 11.38 | 16.23 | 80.23 |
| NCE | 2.02 | 2.48 | 1.97 | 13.00 | 2.00 | 84.28 | 6.21 | 0.57 | 10.90 | 2.93 | 16.23 | 14.08 |

## J    EXAMPLE SOLUTIONS

This section provides the routing examples. Fig. A.3 shows the solution of Rat99-2 computed by LKH-3 and NCE. Figs. A.4 and A.5 shows the solution of a FMDVRP and MDVRP instance computed by OR-Tools and NCE.

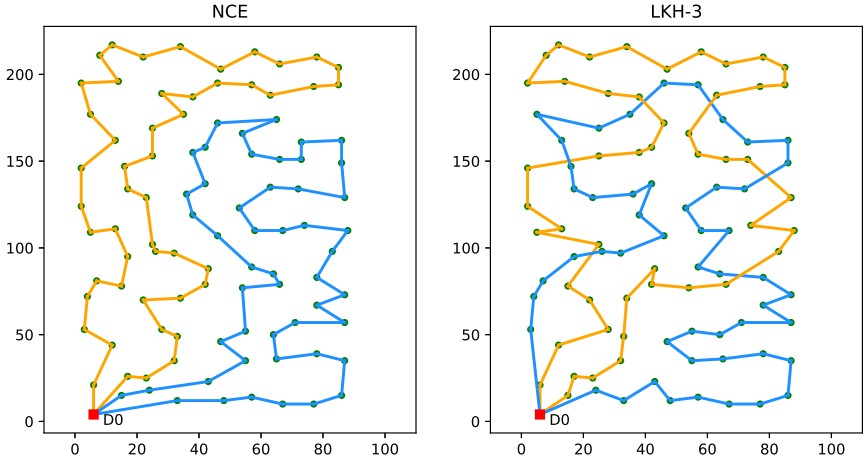

Figure A.3: Rat99-2 solutions computed by NCE and LKH-3

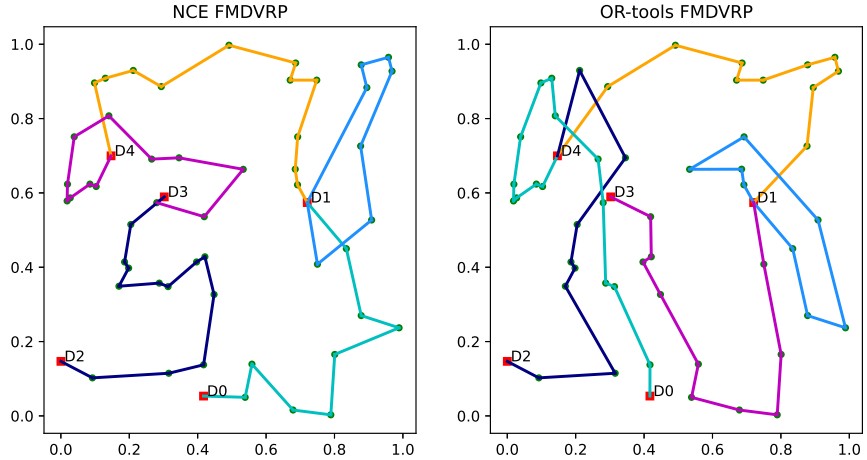

Figure A.4: FMDVRP solutions computed by NCE and OR-tools

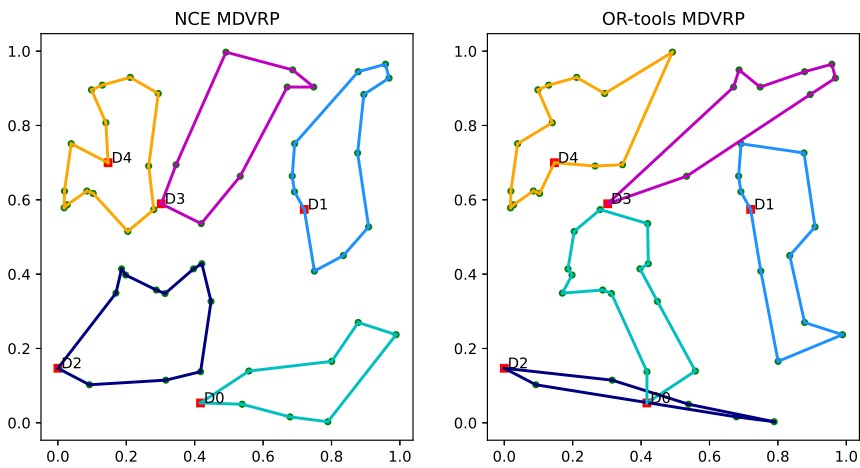

Figure A.5: MDVRP solutions computed by NCE and OR-tools

## K   CASE STUDY: PLANNING PATH FOR STERILIZING ROBOTS

We applied the trained NCE to solve a real-world application, the task of deriving the cooperative path for multiple sterilizing robots to sterilize the open space of a building in the minimum time. Here, the task nodes correspond to the spatial grids composing the entire building floor area, as shown in Fig. A.6. It consists of a total of 500 grids, of which 25 are obstacle areas (gray grids) that a robot cannot move into, and 99 are contaminated areas (orange grids) a robots need to sterilize. We assume the traveling time between the two adjacent grids is 1, the service time for the general area (green grids) is 1, and 2 for the contaminated areas. The upper left corner is the starting point. We formulate finding the cooperative paths for multiple robots to minimize the operation time as the min-max mTSP problem.

Fig. A.7 shows the path derived from NCE when there are 1 to 5 robots, and Table A.15 shows the time and objective value of solving the problem for each algorithm. The results show that the NCE algorithm trained using a synthetic dataset can produce an efficient cooperative path for multiple robots without retraining or fine-tuning. NCE reduces more than 10% of the makespan compared to Google OR-tool and 20% to Greedy+TSP when the number of robots is 5.

Note that the distribution of tasks is far from uniform distribution; each grid is clustered, and each cluster is separated due to the unique floor plan of a building. Thus, the experiment results verify the generalization capability of NCE to problem instances generated from a different distribution.

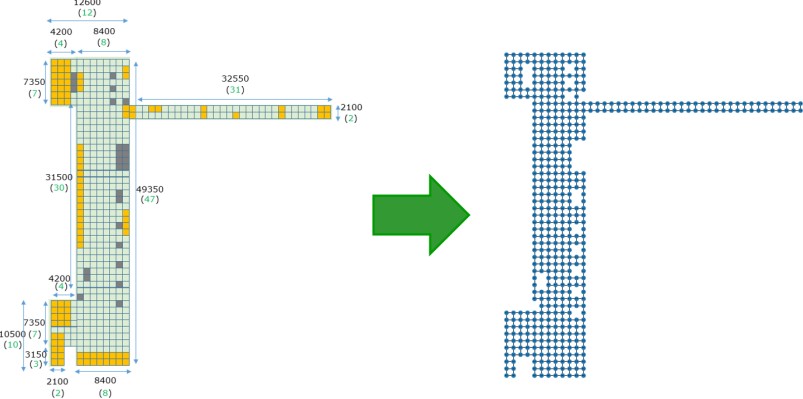

Figure A.6: Grid representation for the open space of the target building

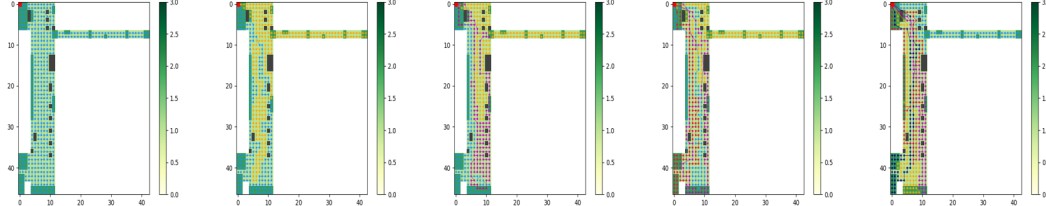

Figure A.7: NCE planned trajectories of the robots under different number of robots

Table A.15: Case study results

| $N_v (\rightarrow)$ Method | 1 Cost | 1 Gap | 1 Time | 2 Cost | 2 Gap | 2 Time | 3 Cost | 3 Gap | 3 Time | 4 Cost | 4 Gap | 4 Time | 5 Cost | 5 Gap | 5 Time |
|---|---|---|---|---|---|---|---|---|---|---|---|---|---|---|---|
| Greedy | 1122 | 6.45 | 0.18 | 631 | 19.06 | 0.21 | 458 | 25.48 | 0.24 | 362 | 24.83 | 0.28 | 306 | 24.90 | 0.29 |
| Greedy + TSP | 1054 | 0.00 | 30.61 | 591 | 11.51 | 14.33 | 432 | 18.36 | 9.75 | 349 | 20.34 | 5.83 | 294 | 20.00 | 4.51 |
| OR-tools | 1066 | 1.14 | 553.2 | 573 | 8.11 | 1624.6 | 383 | 4.93 | 1554.7 | 306 | 5.52 | 2052.5 | 269 | 9.80 | 1043.8 |
| NCE | **1054** | 0.00 | 30.6 | **530** | 0.00 | 450.7 | **365** | 0.00 | 361.7 | **290** | 0.00 | 383.8 | **245** | 0.00 | 233.2 |

## L    BALANCE BETWEEN PERFORMANCE AND COMPUTATION TIME

Because NCE is an improvement heuristic, the more computation time is used, the better outcome will be produced. Thus, we investigate how the performance varies with the allowed computation time. We sampled 100 instances of FMDVRP $(N_c, N_d, N_v)$=(60, 5, 5) and computed the average cost and computation time of NCE and baselines on these problem instances.

Fig. A.8 depicts the relationships between "Run time (computational cost) vs. Cost curve" in the form of a Pareto curve for the case of FMDVRP $(N_c, N_d, N_v)$=(60, 5, 5). The result shows how the performance of NCE improves with more computation time, and this trend is always superior to another baseline.

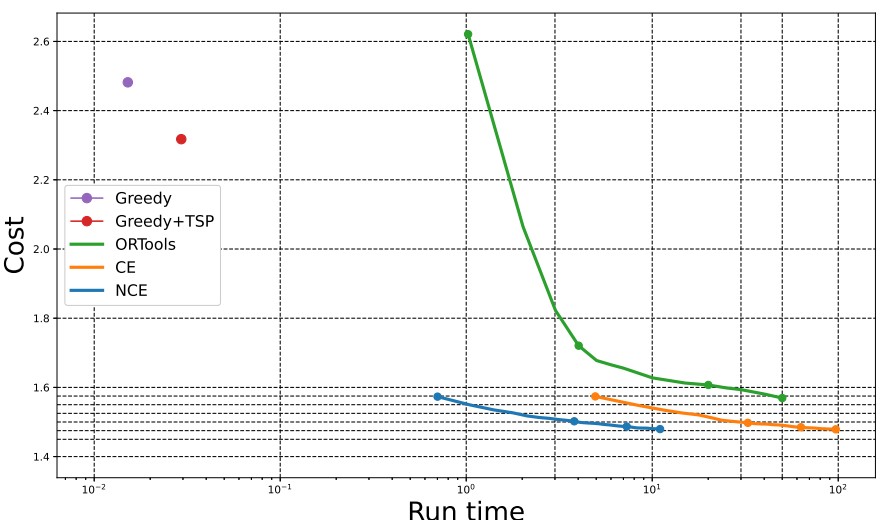

Figure A.8: Performance vs Computation time curve

# M    GENERALIZATION TO DIFFERENT DISTRIBUTION

We investigate the generalization capability of NCE to the problem instances generated from different distributions. We have employed the NCE trained by FMDVRP random instances to the problems generated by the different distributions introduced in (Bi et al., 2022). We used cluster, expansion, explosion, grid, implosion, and mixed distribution as unseen distributions for testing. We followed the definition in the (Bossek et al., 2019; Jiang et al., 2022; Bi et al., 2022), and the data were taken from the (Bi et al., 2022).

Table A.16 summarizes the experiment result. As in the case of uniform distribution, NCE shows leading performance while being faster than OR-Tools and CE. Note that we did not retrain our NCE algorithm but just employed the already trained existing NCE algorithm to new distribution tasks. From the results, we can conclude that the learned $f_\theta$ is generalizable to the different distributions other than the uniform. It again highlights the effectiveness of learning fundamental operators when solving the VRP families.

Table A.16: **Results from various distributions**

| $N_c, N_d, N_v(\rightarrow)$ | | | | | FMDVRP (46,5,5) | | | | |
|---|---|---|---|---|---|---|---|---|---|
| Distribution | | *Cluster* | | | *Expansion* | | | *Explosion* | |
| Method | Cost | Gap(%) | Time(sec.) | Cost | Gap(%) | Time(sec.) | Cost | Gap(%) | Time(sec.) |
| OR-tools | 0.92 | 4.55 | 1.92 | 1.25 | 6.90 | 1.75 | 1.33 | 9.92 | 1.75 |
| Greedy | 1.32 | 50.00 | 0.01 | 1.94 | 67.24 | 0.01 | 1.96 | 61.98 | 0.01 |
| Greedy + TSP | 1.25 | 42.05 | 0.02 | 1.82 | 56.90 | 0.02 | 1.86 | 53.72 | 0.02 |
| CE | 0.88 | 0.00 | 5.82 | 1.16 | 0.00 | 6.50 | 1.39 | 0.00 | 6.39 |
| NCE | **0.88** | 0.00 | 1.51 | **1.16** | 0.00 | 1.39 | **1.21** | 0.00 | 1.64 |
| Distribution | | *Grid* | | | *Implosion* | | | *Mixed* | |
| Method | Cost | Gap(%) | Time(sec.) | Cost | Gap(%) | Time(sec.) | Cost | Gap(%) | Time(sec.) |
| OR-tools | 1.51 | 7.86 | 1.79 | 1.52 | 9.35 | 1.79 | 1.36 | 7.09 | 1.91 |
| Greedy | 2.27 | 62.14 | 0.01 | 2.20 | 58.27 | 0.01 | 2.06 | 62.20 | 0.01 |
| Greedy + TSP | 2.16 | 54.29 | 0.02 | 2.10 | 51.08 | 0.02 | 1.95 | 53.54 | 0.02 |
| CE | 1.41 | 0.71 | 6.07 | 1.39 | 0.00 | 6.34 | 3.03 | 0.00 | 7.41 |
| NCE | **1.40** | 0.00 | 1.77 | **1.39** | 0.00 | 1.84 | **1.27** | 0.00 | 1.90 |
| $N_c, N_d, N_v(\rightarrow)$ | | | | | FMDVRP (94,7,7) | | | | |
| Distribution | | *Cluster* | | | *Expansion* | | | *Explosion* | |
| Method | Cost | Gap(%) | Time(sec.) | Cost | Gap(%) | Time(sec.) | Cost | Gap(%) | Time(sec.) |
| OR-tools | 0.91 | 5.81 | 24.86 | 1.16 | 7.41 | 23.37 | 1.21 | 10.00 | 24.25 |
| Greedy | 1.41 | 63.95 | 0.04 | 1.92 | 77.78 | 0.04 | 1.97 | 79.09 | 0.04 |
| Greedy + TSP | 1.35 | 56.98 | 0.08 | 1.81 | 67.59 | 0.07 | 1.84 | 67.27 | 0.06 |
| CE | 0.86 | 0.00 | 105.66 | 1.08 | 0.00 | 104.06 | 1.10 | 0.00 | 105.82 |
| NCE | **0.86** | 0.00 | 9.11 | **1.08** | 0.00 | 8.60 | **1.10** | 0.00 | 9.70 |
| Distribution | | *Grid* | | | *Implosion* | | | *Mixed* | |
| Method | Cost | Gap(%) | Time(sec.) | Cost | Gap(%) | Time(sec.) | Cost | Gap(%) | Time(sec.) |
| OR-tools | 1.50 | 9.49 | 25.14 | 1.49 | 11.19 | 24.13 | 1.37 | 7.87 | 27.94 |
| Greedy | 2.38 | 73.72 | 0.04 | 2.35 | 75.37 | 0.04 | 2.26 | 77.95 | 0.04 |
| Greedy + TSP | 2.24 | 63.50 | 0.06 | 2.23 | 66.42 | 0.06 | 2.15 | 69.29 | 0.07 |
| CE | 1.37 | 0.00 | 98.27 | 1.34 | 0.00 | 98.55 | 1.27 | 0.00 | 148.73 |
| NCE | **1.37** | 0.00 | 9.24 | **1.34** | 0.00 | 9.49 | **1.27** | 0.00 | 10.11 |

