# OpenReview forum: "Learning to CROSS exchange to solve min-max vehicle routing problems"
_ICLR.cc/2023/Conference — ICLR 2023 poster_

### Official Review · Reviewer_vbx7 · 2022-10-23

**Confidence:** 4
**Clarity, Quality, Novelty And Reproducibility:** The clarity and quality are both fine…
**Correctness:** 3
**Technical Novelty And Significance:** 2
**Empirical Novelty And Significance:** 2
**Recommendation:** 5

**Strength And Weaknesses:**

Strength:
- The proposed NeuroCROSS is reasonable to reduce the complexity of CE operation.
- The experimental results show the proposed method has favorable performance on both medium-sized and large-sized instances and can be generalized to similar problems.

Weaknesses:

- There has been many works trying to learn heurisitcs (using reinforcement learning or supervised learning) for combinatorial optimization. This work is just another piece of standard implemention, using supervised learning. Indeed I did not find any new insight.

- The design to search two starting nodes by prediction is too heuristic. There are four nodes to search in the CE operation, but the reason why to select the two starting nodes for prediction is not well-explained.

- The generalization performance on different instance-generation distributions is not validated in the paper. The customer nodes of all the training and testing instances are generated by uniform distribution. The authors should verify the generalization performance on other distributions such as explosion, implosion, and so on [1].

[1] Jiang Y, Wu Y, Cao Z, et al. Learning to Solve Routing Problems via Distributionally Robust Optimization[C]. 36th AAAI Conference on Artificial Intelligence (AAAI), Virtual Event, 2022, 9786--9794.


**Summary Of The Paper:**

The paper proposed a new learning-based heuristic, named NeuroCROSS, to solve several routing problems. Unlike the previously learned intra-operations like 2-opt and swap, the proposed NeuroCROSS is an inter-operation between two routes, which requires different learning task formulation and neural network design. The proposed method utilizes a supervised learning formulation to predict the best cost-decrement based on two starting nodes and thus reduce the complexity. Several numerical experiments are conducted, and the proposed method outperforms meta-heuristic baselines and neural baselines.


**Summary Of The Review:**

This paper improves the complexity of a fundamental CE operation by using supervised learning to predict the cost decrement. The heuristic solvers can utilize the proposed NeuroCROSS to reduce time consumption. This work is not bad, but the novelty is limited, making it not reach the standard of a top venue. Furthermore, the proposed method is a little too heuristic and the experiments also need to be enhanced.

---

> ### Author Response · Authors · 2022-11-13
> **Reply for the Reviewer vbx7 (1)**
>
> Thank you for your valuable input. We share our reply to your review.
>
> **About weakness 1: There has been many works trying to learn heuristics (using reinforcement learning or supervised learning) for combinatorial optimization. This work is just another piece of standard implementation, using supervised learning. Indeed I did not find any new insight.**
>
> As far as we know, learning-based heuristic methods for combinatorial optimization can be categorized into (1) construction heuristics and (2) improving heuristics. In addition, depending on the learning method, these algorithms can also be categorized into (1) reinforcement learning based and (2) imitation learning (supervised learning based). NCE belongs to improvement heuristics based on imitation learning.
>
>
> Although NCE belongs to one of such categories, NCE has unique novelties that other methods in the same class do not possess. For example, while other methods do not have transferability to other problem classes, NCE learns a fundamental and universal operator to solve various complex min-max VRPs without retraining for each type of VRPs. In addition, NCE does not have to solve and evaluate VRP instances during training but just needs to use the local search results, which facilitates training.
>
>
> Although the methodology seems too simple to be technically novel, NCE contributes to developing general-purpose algorithms by learning key local operator common to many problems and proposing strategies to reduce computation time.
>
>
> The following table summarizes the generalization capability of NCE and other baselines:
>
> |   |Target problem classes  |Generalizable to problems of different sizes?  |Generalizable to problems of different classes?  |
> |--|--|--|--|
> |An Efficient Graph Convolutional Network Technique for the TSP (Joshi, 2019) [1]| TSP | X|X|
> |NLNS (Hottung & Tierney, 2019) [2] | TSP, CVRP, mTSP |X |X  |
> |NeuroRewriter (Chen & Tian, 2019) [3]| JSSP, CVRP|X |X |
> | Learn2OPT (da Costa et al., 2021) [4]| TSP, CVRP|X | X |
> |NeuroLKH (Liang Xin, 2021) [5] |CVRP, VRPTW, PDP  | O| X|
> |Learning to Delegate (Sirui Li, 2021) [6]|CVRP, VRPTW, PDP  | O| X|
> Learning improvement heuristic (Yaoxin Wu, 2021) [7]|TSP, CVRP|X | X |
> |NCE  | Min-max FMDVRP, Min-max MDVRP, Min-max mTSP,Min-max CVRP  | O |O  |
>
> Reference:
>
> [1]: Chaitanya K Joshi, Thomas Laurent, and Xavier Bresson. An efficient graph convolutional network
> technique for the travelling salesman problem. arXiv preprint arXiv:1906.01227, 2019.
>
> [2]: Andr´e Hottung and Kevin Tierney. Neural large neighborhood search for the capacitated vehicle
> routing problem. arXiv preprint arXiv:1911.09539, 2019.
>
> [3]: Xinyun Chen and Yuandong Tian. Learning to perform local rewriting for combinatorial optimization.
> In Advances in Neural Information Processing Systems, pp. 6281–6292, 2019.
>
> [4]: Paulo da Costa, Jason Rhuggenaath, Yingqian Zhang, Alp Akcay, and Uzay Kaymak. Learning
> 2-opt heuristics for routing problems via deep reinforcement learning. SN Computer Science, 2
> (5):1–16, 2021
>
> [5]: Liang Xin, Wen Song, Zhiguang Cao, and Jie Zhang. Neurolkh: Combining deep learning model
> with lin-kernighan-helsgaun heuristic for solving the traveling salesman problem. Advances in
> Neural Information Processing Systems, 34, 2021b.
>
> [6]: Sirui Li, Zhongxia Yan, and Cathy Wu. Learning to delegate for large-scale vehicle routing. Advances
> in Neural Information Processing Systems, 34, 2021.
>
> [7]: Yaoxin Wu, Wen Song, Zhiguang Cao, Jie Zhang, and Andrew Lim. Learning improvement heuristics
> for solving routing problem. IEEE Transactions on Neural Networks and Learning Systems,
> 2021.

---

> ### Author Response · Authors · 2022-11-13
> **Reply for the Reviewer vbx7 (2)**
>
> **About weakness 2: The design to search two starting nodes by prediction is too heuristic. There are four nodes to search in the CE operation, but the reason why to select the two starting nodes for prediction is not well-explained.**
>
> The principal idea behind the cost-decrement prediction function is amortizing the exhaustive search after selecting the initial nodes.
>
> Let denote $a_1$ and $b_1$ as the starting and ending node for the sub route of trajectory $τ_1$, respectively.
>
> Let denote $a_2$ and $b_2$ as the starting and ending node for the sub route of trajectory $τ_2$, respectively.
>
> There are several ways to construct the cost decrement perdition models. We list out all the possible ways to construct the cost decrement prediction model $f_θ$ as follows:
>
> **(The number of initial node section = 1)**
>
> $y(a_1; \tau_1, \tau_2) = \max_{(a_2, b_1, b_2)}\left(C(a_1,b_1,a_2,b_2; \tau_1, \tau_2)-C(\tau_1, \tau_2)\right) \approx f_\theta(a_1; \tau_1, \tau_2)$
>
> $y(a_2; \tau_1, \tau_2) = \max_{(a_1, b_1, b_2)}\left(C(a_1,b_1,a_2,b_2; \tau_1, \tau_2)-C(\tau_1, \tau_2)\right) \approx f_\theta(a_2; \tau_1, \tau_2)$
>
> $y(b_1; \tau_1, \tau_2) = \max_{(a_1, a_2, b_2)}\left(C(a_1,b_1,a_2,b_2; \tau_1, \tau_2)-C(\tau_1, \tau_2)\right) \approx f_\theta(b_1; \tau_1, \tau_2)$
>
> $y(b_2; \tau_1, \tau_2) = \max_{(a_1, a_2, b_1)}\left(C(a_1,b_1,a_2,b_2; \tau_1, \tau_2)-C(\tau_1, \tau_2)\right) \approx f_\theta(b_2; \tau_1, \tau_2)$
>
> The computation cost for NCE operation would be $O(N) + KO(N^3)$, where $K$ is the number of candidates set.
>
> **(The number of initial node section = 2)**
>
> $y(a_1, a_2; \tau_1, \tau_2) = \max_{b_1, b_2}\left(C(a_1,b_1,a_2,b_2; \tau_1, \tau_2)-C(\tau_1, \tau_2)\right) \approx f_\theta(a_1, a_2; \tau_1, \tau_2)$
>
> $y(b_1,  b_2; \tau_1,  \tau_2) = \max_{a_1, a_2}\left(C(a_1,b_1,a_2,b_2; \tau_1, \tau_2)-C(\tau_1, \tau_2)\right) \approx f_\theta(b_1, b_2; \tau_1, \tau_2)$
>
>
> $y(a_1, b_2; \tau_1, \tau_2) = \max_{a_2, b_1}\left(C(a_1,b_1,a_2,b_2; \tau_1, \tau_2)-C(\tau_1, \tau_2)\right) \approx f_\theta(a_1, b_2; \tau_1, \tau_2)$
>
> $y(a_2, b_1; \tau_1, \tau_2) = \max_{a_1, b_2}\left(C(a_1,b_1,a_2,b_2; \tau_1, \tau_2)-C(\tau_1, \tau_2)\right) \approx f_\theta(a_2, b_1; \tau_1, \tau_2)$
>
> $y(a_1, b_1; \tau_1, \tau_2) = \max_{a_2, b_2}\left(C(a_1,b_1,a_2,b_2; \tau_1, \tau_2)-C(\tau_1, \tau_2)\right) \approx f_\theta(a_1, b_1; \tau_1, \tau_2)$
>
> $y(a_2, b_2; \tau_1, \tau_2) = \max_{a_1, b_1}\left(C(a_1,b_1,a_2,b_2; \tau_1, \tau_2)-C(\tau_1, \tau_2)\right) \approx f_\theta(a_2, b_2; \tau_1, \tau_2)$
>
> The computation cost for NCE operation would be $O(N^2) + KO(N^2)$, where $K$ is the number of candidates set.
>
> **(The number of initial node section = 3)**
>
> $y(a_1, a_2,b_1; \tau_1, \tau_2) = \max_{b_2}\left(C(a_1,b_1,a_2,b_2; \tau_1, \tau_2)-C(\tau_1, \tau_2)\right) \approx f_\theta(a_1, a_2,b_1; \tau_1, \tau_2)$
>
> $y(a_1, a_2,b_2; \tau_1, \tau_2) = \max_{b_1}\left(C(a_1,b_1,a_2,b_2; \tau_1, \tau_2)-C(\tau_1, \tau_2)\right) \approx f_\theta(a_1, a_2,b_2; \tau_1, \tau_2)$
>
> $y(a_1, b_1,b_2; \tau_1, \tau_2) = \max_{a_2}\left(C(a_1,b_1,a_2,b_2; \tau_1, \tau_2)-C(\tau_1, \tau_2)\right) \approx f_\theta(a_1, b_1,b_2; \tau_1, \tau_2)$
>
> $y(a_2, b_1,b_2; \tau_1, \tau_2) = \max_{a_1}\left(C(a_1,b_1,a_2,b_2; \tau_1, \tau_2)-C(\tau_1, \tau_2)\right) \approx f_\theta(a_2, b_1,b_2; \tau_1, \tau_2)$
>
> The computation cost for NCE operation would be $O(N^3) + KO(N)$, where $K$ is the number of candidates set.
>
> **(The number of initial node section = 4)**
>
> $y(a_1, a_2,b_1,b_2; \tau_1, \tau_2) = C(a_1,b_1,a_2,b_2; \tau_1, \tau_2)-C(\tau_1, \tau_2)$
>
> The computation cost for NCE operation would be $O(N^4) )$.
>
> Therefore, using the two nodes as inputs for  $f_θ$ is the most beneficial in terms of computational cost. Also note that there are a total six possible ways to choose two nodes to be used for $f_θ$.
>
> These six options can be further categorized into three due to symmetricity:
>
> (1) choosing two nodes from the same side of the two trajectories.
>
> (2) choosing two nodes from different sides of two trajectory.
>
> (3) choosing the two nodes from the same trajectory.
>
> Among the three options, we choose (1) because choosing $(a_1,a_2)$ directly induces the removal of edges: $(a_1,a_1+1)$, $(a_2,a_2+1)$ and the insertion of edges: $(a_1,a_2+1)$ and $(a_2,a_1+1)$, as shown in **Figure 2** in the paper. Because the edge to be removed and the edge to be added are clear due to the selection of two nodes, it is easy to predict the cost decrement. If we use (2), it is hard to know what edges will be inserted and removed, thus the cost decrement is hard to be estimated. Option (3) is not effective neither because it cannot consider the interaction between the selected two trajectories.
>
> For these reasons, we decided to use $f_θ (a_1,a_2,τ_1,τ_2)$ using the first two nodes as an input.

---

> ### Author Response · Authors · 2022-11-13
> **Reply for the Reviewer vbx7 (3)**
>
> **About weakness 3: The generalization performance on different instance-generation distributions is not validated in the paper. The customer nodes of all the training and testing instances are generated by uniform distribution. The authors should verify the generalization performance on other distributions such as explosion, implosion, and so on [1].**
>
> [1]: Jiang Y, Wu Y, Cao Z, et al. Learning to Solve Routing Problems via Distributionally Robust Optimization[C]. 36th AAAI Conference on Artificial Intelligence (AAAI), Virtual Event, 2022, 9786--9794.
>
> We believe the biggest advantage of NCE is generalization performance. We use a single $f_θ (·)$ that is trained using data obtained from random min-max FMDVRP instances for all experiments. We found that $f_θ (·)$ effectively solves the three special cases (i.e., min-max MDVRP, min-max mTSP, and min-max CVRP) without retraining or fine tuning, proving the effectiveness of NCE as a universal operator for different classes of VRPs. In addition, the experiment results proving its generalization over the different distribution has already been provided in the appendix of the manuscript (CVRPlib, and mTSPlib performance evaluation results).
>
> The following three aspects prove an efficient generalization capability of NCE. Respecting the reviewer’s comment, we conducted two additional experiments:
>
> **(Existing experiment result: non-uniform benchmark data set)**
>
> We have already applied the learned NCE trained from the uniform FMDVRP instances to solve mTSP instances from mTSPlib[2] and CVRP instances from CVRPlib[3]. The instances from these two libraries have task distribution far from the uniform distribution and are considered practical tasks. The result on mTSPlib and CVRPlib are provided in **Table A.4 and Table A.6 in Appendix**, which show that NCE still outperforms other baselines.
>
> [2] mTSPLib. URL https://profs.info.uaic.ro/~mtsplib/MinMaxMTSP/
>
> [3] CVRPLib. URL http://vrp.galgos.inf.puc-rio.br/index.php/en/
>
> **(New experiment result: employment to real-world applications)**
>
> We have recently employed the NCE algorithm to solve real-world applications. For example, we have included the cooperative planning of multiple sterilizing robots in a building. Here, the task nodes correspond to the spatial grids composing the entire building floor area. Thus, the distribution of tasks is far from uniform distribution; each grid is clustered, and each cluster is separated due to the unique floor plan of a building. The results show that the NCE algorithm trained using a synthetic dataset can produce an efficient cooperative path for multiple robots without retraining or fine-tuning. Furthermore, it reduces more than 10% of the makespan compared to Google OR-tool and 20% to Greedy+TSP. Please see **Appendix K** for the details of the task and the results.
>
> **(New experiment result: employment to the distributions (explosion, implosion, and so) used in [1])**
>
> As suggested by the reviewer, we have employed the NCE trained by FMDVRP random instances to the problems generated by the different distributions introduced in [1]. Note that we did not retrain our NCE algorithm but just employed the already trained existing NCE algorithm to new distribution tasks. In **Appendix M** of the revised manuscript, we have included the result of it. The following table summarizes the new experiment results in FMDVRP $(N_c, N_d, N_v)$=(94, 7, 7) instances. The results showing that NCE still outperforms other baseline algorithms in solving FMDVRP instances generated from various distribution other than the uniform:
>
> |     Distribution    |     Cluster    |  |     Expansion    |  |     Explosion    |  |     Grid    |  |     Implosion    |  |     Mixed    |  |
> |---|---|---|---|---|---|---|---|---|---|---|---|---|
> |         |     Cost       |     Time    |     Cost       |     Time    |     Cost       |     Time    |     Cost       |     Time    |     Cost       |     Time    |     Cost       |     Time    |
> |     Greedy    |     1.41    |     0.04    |     1.92    |     0.04    |     1.97    |     0.04    |     2.38    |     0.04    |     2.35    |     0.04    |     2.26    |     0.04    |
> |     Greedy   + TSP    |     1.35    |     0.08    |     1.81    |     0.07    |     1.84    |     0.06    |     2.24    |     0.06    |     2.23    |     0.06    |     2.15    |     0.07    |
> |     OR-tool    |     0.91    |     24.86    |     1.16    |     23.37    |     1.21    |     24.25    |     1.5    |     25.14    |     1.49    |     24.13    |     1.37    |     27.94    |
> |     CE    |     0.86    |     105.66    |     1.08    |     104.06    |     1.1    |     105.82    |     1.37    |     98.27    |     1.34    |     98.55    |     1.27    |     148.73    |
> |     NCE    |     0.86    |     9.11    |     1.08    |     8.6    |     1.1    |     9.7    |     1.37    |     9.24    |     1.34    |     9.49    |     1.27    |     10.11    |

---

### Official Review · Reviewer_LBRJ · 2022-10-25

**Confidence:** 4
**Correctness:** 4
**Technical Novelty And Significance:** 3
**Empirical Novelty And Significance:** 3
**Recommendation:** 8

**Clarity, Quality, Novelty And Reproducibility:**

Very well written and structure, a pleasure to read.
The proposed methods are novel.
Reproducibility is medium (experiments are clearly described, but no code is
provided).


**Strength And Weaknesses:**

s1. very simple idea.
s2. learnable in a supervised way.
s3. applicable to different problem variants.

w1. results are hard to compare due to two objectives (cost & runtime)
w2. focus only on min-max / makespan problems, without results
  for more intensily researched vanilla CVRPs.


**Summary Of The Paper:**

The paper addresses the problem of vehicle routing, esp.
learning local search neighborhoods. The authors investigate
esp. the cross operator that swaps two subsequences between
two tours. The authors propose a hybrid neighborhood that
uses a model that given two tours predicts and two candidate
starting points of the two segments predicts the cost savings,
and then searches through all possible end points to find good
crossings. In experiments they compare their method against
several baselines from operations research and machine learning
across different variants of the VRP problem.



**Summary Of The Review:**

I see two potential issues:
w1. results are hard to compare due to two objectives (cost & runtime)

  For mTSPs (tab. 4) baseline DAN is way faster than the proposed method.
  To compare just the cost for different runtimes, is not so meaningful.
  Cannot DAN also make good use of the remaining time, with random
  samplings (as done in table 5 for baseline HM) or with post-processing
  of the solutions with a heuristic local seach?

w2. focus only on min-max / makespan problems, without results
  for more intensily researched vanilla CVRPs.

  Also covering the less researched problems is good, of course.
  But cannot you also compare your method on vanilla CVRPs where
  you would have more learned local search baselines and other
  ML baselines? -- If not, is there anything special about the min-max
  VRP problems that your method is designed to model?

---

> ### Author Response · Authors · 2022-11-13
> **Reply for the Reviewer LBRJ**
>
> Thank you for your valuable input. We share our reply to your review.
>
> **About weakness 1: Results are hard to compare due to two objectives (cost & runtime). For mTSPs (tab. 4) baseline DAN is way faster than the proposed method. To compare just the cost for different run times, is not so meaningful. Cannot DAN also make good use of the remaining time, with random samplings (as done in table 5 for baseline HM) or with post-processing of the solutions with a heuristic local search?**
>
> We agree that it is necessary to compare the performance change according to the computation time of the algorithms to compare the performance of algorithms seriously.
>
> We did not implement DAN by ourselves and took the results in Table 4 from their paper directly. We might be able to identify some hyper-parameters that can control the balance between the computation time and solution quality, but we reserve to identify them and modify the DAN algorithm since we cannot be confident that our implementation produces the results correctly. For mTSPlib tasks, DAN is employed with sampling strategies from their paper. Even with the sampling strategy, DAN’s performance is significantly inferior to NCE. This result is provided in **Table A.4 in Appendix C**.
>
> The computation time for DAN and NCE are distinctly different because their solution-finding procedure is entirely different. DAN and ScheduleNet are construction heuristic methods that construct a complete solution through one step of the encoding and decoding process. On the other hand, NCE is improvement heuristics that iteratively update the solution through the local improving operator until it reaches a good solution. In general, improvement heuristic produces a better-quality solution but takes more time than construction heuristics.
>
> **About weakness 2: Focus only on min-max / makespan problems, without results for more intensely researched vanilla CVRPs. Also covering the less researched problems is good, of course. But cannot you also compare your method on vanilla CVRPs where you would have more learned local search baselines and other ML baselines? -- If not, is there anything special about the min-max VRP problems that your method is designed to model?**
>
> **(NCE employment to solve min-sum CVRP)**
>
> Cross-exchange can be employed for any cost objective; we also employed NCE to solve min-sum CVRP problems. For min-sum CVRP, we trained the NCE cost decrement model using different cost definitions to minimize the sum of the traveling distance. The min-sum CVRP benchmark results are provided in **Table A.7 in Appendix E**. Although the NCE does not provide the best performance, it derived a solution whose performance is similar to that of other learning-based baselines. This is NCE has unique structure advantageous for solving min-max VRP, which will be explained in the following paragraph.
>
> **(The unique structure designed for min-max VRP)**
>
> We noticed that the inter-operation of CE is especially effective in improving the quality of “min-max VRP” because it can consider the interaction among multiple vehicles and effectively reduce the differences between the traveling distances of all vehicles. Thus, we have designed NCE to solve min-max VRPs, the problem having important practicality but being studied less. The practical importance of min-max VRP is described in the following paragraph.
>
> **(Practical importance of min-max VRP)**
>
> We have noticed that most research focuses on min-sum VRP since most benchmark problems and baseline algorithms are well-established. However, we found several practical problems that aim to minimize the total completion time, for example, various time-critical distributed tasks (e.g., vaccine delivery, grocery delivery).

---

### Official Review · Reviewer_5SXq · 2022-10-25

**Confidence:** 4
**Correctness:** 4
**Technical Novelty And Significance:** 3
**Empirical Novelty And Significance:** 3
**Recommendation:** 8

**Clarity, Quality, Novelty And Reproducibility:**

Clarity: Main ideas of the paper is easy to follow. Technical notations are consistent throughout the document. The introduction succinctly & insightfully describes main ideas, and builds clear expectations on what is going to come next

Quality: The proposed method applies to a very generic class of problems (FMDVRP), many of them are of practical importance. Authors follow a principled methodology, and evaluates across a good range of problems & baselines.

Novelty: The idea of approximating a value/cost function in a heuristic with a neural network model is not particularly new, but the particular idea of splitting the decision to two pieces (choose one ends of the route first, and then choose the other ends exactly) is new; this strategy would generalize to other meta-heuristics.

(Question 1: could we choose (a1, b2) instead of (a1, b1)? Question 2: could we reduce the compute time from O(n^2) to O(n) by choosing one end of one route at a time?)

Reproducibility: Authors mostly use synthetic data. The compute resource required is not very high. I didn't see whether the code is going to be shared. It would take moderate effort to re-implement the approach.

**Details Of Ethics Concerns:**

Neural combinatorial optimization methods will introduce some bias on real-world applications they solve, but it is yet unclear what kind of harm they will cause.

**Strength And Weaknesses:**

Strengths:

Authors demonstrate a significant benefit against strong neural & non-neural baselines across several classes of VRPs. The consistency of empirical benefit is quite impressive. Authors also provide a good breadth of ablation studies to justify most components of the proposed method.

Authors build upon well-established methodology from GNN literature, making the method principled and also opening up future adoption of innovations on GNNs.

Authors make a novel connection between a well-established meta-heuristic (CE) and neural combinatorial optimization. The idea of connection (estimating the value function of the heuristic) is interesting on its own.

Weakness:

NCE is only evaluated on a single configuration in Table 1, 2, 3, 4, 5. Since NCE is much more expensive than Greedy/Greedy+TSP, it would be interesting to see the entire compute cost vs. gap curve for both these baselines and the proposed NCE method.

Evaluation is only on synthetic datasets, which limits our understanding on how the method would perform in real-world problems.

**Summary Of The Paper:**

Authors propose a method of accelerating cross exchange (CE) heuristic by approximating the associated "value" function with a neural network. Authors adapt standard GNN architectures to approximate the value function. The method applies to a general class of min-max flexibile multi-depot vehicle routing planning (FMDVRP) problems. The proposed method is evaluated on several classes of FMDVRP problems, outperforming strong neural and non-neural baselines, either in solution quality or in computation.



**Summary Of The Review:**

Generic, principled approach that delivers good advantage over strong baselines on synthetic experiments.

---

> ### Author Response · Authors · 2022-11-13
> **Reply For the  reviewer 5SXq (1)**
>
> Thank you for your valuable input. We share our reply to your review.
>
> **About Weakness 1: NCE is only evaluated on a single configuration in Table 1, 2, 3, 4, 5. Since NCE is much more expensive than Greedy/Greedy+TSP, it would be interesting to see the entire compute cost vs. gap curve for both these baselines and the proposed NCE method.**
>
> Because NCE is an improvement heuristic, the more computation time is used, the better outcome will be produced. On the other hand, constructive heuristics require much shorter computational time because these methods construct a complete solution in a single decoding process.
>
> We have additionally investigated how the performance varies with the allowed computation time. We sampled 100 instances of FMDVRP $(N_c, N_d, N_v)$=(60, 5, 5) and computed the average cost and computation time of NCE and baselines on these problem instances.
>
> **In Appendix L** of the revised manuscript, we have included **Figure A.8** depicting the relationships between “Run time (computational cost) vs. Cost curve” in the form of a Pareto curve. The result shows how the performance of NCE improves with more computation time, and this trend is always superior to another baseline.
>
> **About Weakness 2: Evaluation is only on synthetic datasets, which limits our understanding on how the method would perform in real-world problems.**
>
> We have recently employed the NCE algorithm to solve real-world applications. For example, we have included the cooperative planning of multiple sterilizing robots in a building. Here, the task nodes correspond to the spatial grids composing the entire building floor area. Thus, the distribution of tasks is far from uniform distribution; each grid is clustered, and each cluster is separated due to the unique floor plan of a building. The results show that the NCE algorithm trained using a synthetic dataset sampled from a uniform distribution of FMDVRP can produce an efficient cooperative path for multiple robots without retraining or fine-tuning. Furthermore, it reduces more than 10% of the makespan compared to Google OR-tool and 20% to Greedy+TSP. Please see **Appendix K** for the details of the task and the results.
>
> **About Question1:  could we choose (a1, b2) instead of (a1, b1)? Question 2: could we reduce the compute time from O(n^2) to O(n) by choosing one end of one route at a time?**
>
> The principal idea behind the cost-decrement prediction function is amortizing the exhaustive search after selecting the initial nodes.
>
> Let denote $a_1 $ and $b_1$ as the starting and ending node for the sub route of trajectory $τ_1$, respectively.
>
> Let denote $a_2$ and $b_2$ as the starting and ending node for the sub route of trajectory $τ_2$, respectively.
>
> There are a total six ways to choose two inputs for the cost decrement prediction model $f_θ$ as follows:
> (1) choosing two nodes from the same side of the two trajectories,
> $y(a_1, a_2; \tau_1, \tau_2) = \max_{b_1, b_2}\left(C(a_1,b_1,a_2,b_2; \tau_1, \tau_2)-C(\tau_1, \tau_2)\right) \approx f_\theta(a_1, a_2; \tau_1, \tau_2)$
>
> $y(b_1,  b_2; \tau_1,  \tau_2) = \max_{a_1, a_2}\left(C(a_1,b_1,a_2,b_2; \tau_1, \tau_2)-C(\tau_1, \tau_2)\right) \approx f_\theta(b_1, b_2; \tau_1, \tau_2)$
>
> (2) choosing two nodes from different sides of two trajectory, and
>
> $y(a_1, b_2; \tau_1, \tau_2) = \max_{a_2, b_1}\left(C(a_1,b_1,a_2,b_2; \tau_1, \tau_2)-C(\tau_1, \tau_2)\right) \approx f_\theta(a_1, b_2; \tau_1, \tau_2)$
>
> $y(a_2, b_1; \tau_1, \tau_2) = \max_{a_1, b_2}\left(C(a_1,b_1,a_2,b_2; \tau_1, \tau_2)-C(\tau_1, \tau_2)\right) \approx f_\theta(a_2, b_1; \tau_1, \tau_2)$
>
> (3) choosing the two nodes from the same trajectory.
>
> $y(a_1, b_1; \tau_1, \tau_2) = \max_{a_2, b_2}\left(C(a_1,b_1,a_2,b_2; \tau_1, \tau_2)-C(\tau_1, \tau_2)\right) \approx f_\theta(a_1, b_1; \tau_1, \tau_2)$
>
> $y(a_2, b_2; \tau_1, \tau_2) = \max_{a_1, b_1}\left(C(a_1,b_1,a_2,b_2; \tau_1, \tau_2)-C(\tau_1, \tau_2)\right) \approx f_\theta(a_2, b_2; \tau_1, \tau_2)$
>
> Among the three options, we choose (1) because choosing $(a_1,a_2)$ directly induces the removal of edges: $(a_1,a_1+1), (a_2,a_2+1) $and the insertion of edges: $(a_1,a_2+1)$ and $(a_2,a_1+1)$, as shown in **Figure 2** of the paper. Because the edge to be removed and the edge to be added are clear due to the selection of two nodes, it is easy to predict the cost decrement. If we use (2), it is hard to know what edges will be inserted and removed, thus the cost decrement is hard to be estimated. Option (3) is not effective neither because it cannot consider the interaction between the selected two trajectories.
>
> For these reasons, we decided to use $f_\theta(a_1, a_2; \tau_1, \tau_2)$  using the first two nodes as an input.

---

> ### Author Response · Authors · 2022-11-13
> **Reply For the reviewer 5SXq (2)**
>
> **About Question2: could we reduce the compute time from O(n^2) to O(n) by choosing one end of one route at a time?**
>
> I think the proposed idea is really interesting. To make sure we understand the reviewer’s comment correctly, we have described the overall procedure of the proposed local search as follows:
>
> If we want to select one of two starting nodes sequentially at a time $(a_1→a_2)$, we need to train the two-cost decrement-prediction model:
>
> $f^1_\theta(a_1, \tau_1, \tau_2)  \approx \max_{a_2, b_1, b_2}\left(C(a_1,b_1,a_2,b_2; \tau_1, \tau_2)-C(\tau_1, \tau_2)\right) = y(a_1; \tau_1, \tau_2) $
>
> $f^2_\theta(a_1, a_2; \tau_1, \tau_2) \approx \max_{b_1, b_2}\left(C(a_1,b_1,a_2,b_2; \tau_1, \tau_2)-C(\tau_1, \tau_2)\right) =  y(a_1, a_2; \tau_1, \tau_2) $
>
> $f_θ^1$ uses a single starting node $a_1$ as an input and amortizes the search process of the rest of three nodes $(a_2,b_1,b_2)$.
> $f_θ^2$ uses two starting nodes $(a_1,a_2)$ as an input and amortizes the search process of the rest of two nodes $(b_1,b_2)$.
>
> Using these two trained models, we can then select sequentially the two starting nodes as:
>
> $a_1^* =$ $argmax_{a_1}$$f^1_\theta(a_1, \tau_1, \tau_2)$
>
> $a_2^* =$ $argmax_{a_2}$$f^1_\theta(a_1^*,a_2, \tau_1, \tau_2)$
>
> Thus, the search cost can be $O(n)$ instead of $O(n^2)$. However, to train $f_θ^1 (a_1,τ_1,τ_2)$, we have to compute the true labels $\max_{a_2, b_1, b_2}C(a_1,b_1,a_2,b_2; \tau_1, \tau_2)-C(\tau_1, \tau_2)$, which incurs the search cost of $O(n^3)$. Thus, the more time will be required to generate the true labels. More importantly, predicting the cost decrement using only a single specified node would be more challenging due to the lack of information given as an input;  amortizing the search over three variables is harder than amortizing the search of two variables.
>
> We believe it will be really meaningful comparing how the computational cost and performance balance varies with the the input-option for the cost-decrement prediction model. We would like leave it as the future research
>
> **Reproducibility: Authors mostly use synthetic data. The compute resource required is not very high. I didn't see whether the code is going to be shared. It would take moderate effort to re-implement the approach**
>
> We are going to release our code if the paper is accepted.

---

### Decision · Program_Chairs · 2023-01-20

**Decision:**

Accept: poster

**Justification For Why Not Higher Score:**

I don't mind if this paper is bumped up to be spotlight :)

**Justification For Why Not Lower Score:**

NA

**Metareview: Summary, Strengths And Weaknesses:**

This paper proposes a new method Neuro CE, a learning-based heuristic method for combinatorial optimization. Reviewers are impressed by the significant benefits compared to both neural and non-neural baselines across several classes of VRPs. Overall two reviewers endorsed the paper enthusiastically, while one reviewer remain relatively conservative. The authors' response provided a thorough explanation for various questions raised, and addressed the novelty concern (which was the major one) satisfactorily & cleverly.

I do think the work has value for the field, given the proposed method learns a general-purpose operator that can solve various complex min-max VRPs without retraining for each type of VRPs. This work may also inspire researchers who seek novel ways to combine ML methods with existing combinatorial optimization algorithms.

I recommend acceptance, and invite authors to incorporate the changes and new experiments in the final version.

Congratulations!



**Note From Pc:**

if the above contains the word "oral" or "spotlight" please see: "oral" presentation means -> notable-top-5% and "spotlight" means -> notable-top-25%. As stated in our emails, we are disassociating presentation type from AC recommendations